# Species Diversity and Distribution Characteristics of *Calonectria* in Five Soil Layers in a *Eucalyptus* Plantation

**DOI:** 10.3390/jof7100857

**Published:** 2021-10-13

**Authors:** LingLing Liu, WenXia Wu, ShuaiFei Chen

**Affiliations:** 1China Eucalypt Research Centre (CERC), Chinese Academy of Forestry (CAF), Zhanjiang 524022, Guangdong Province, China; liulinglingfp@126.com (L.L.); wuwenxia_hainan@126.com (W.W.); 2Nanjing Forestry University (NJFU), Nanjing 210037, Jiangsu Province, China

**Keywords:** fungal ecology, multi-gene phylogeny, plant pathogen, soil-borne fungi, tree disease

## Abstract

The genus *Calonectria* includes pathogens of various agricultural, horticultural, and forestry crops. Species of *Calonectria* are commonly collected from soils, fruits, leaves, stems, and roots. Some species of *Calonectria* isolated from soils are considered as important plant pathogens. Understanding the species diversity and distribution characteristics of *Calonectria* species in different soil layers will help us to clarify their long-term potential harm to plants and their patterns of dissemination. To our knowledge, no systematic research has been conducted concerning the species diversity and distribution characteristics of *Calonectria* in different soil layers. In this study, 1000 soil samples were collected from five soil layers (0–20, 20–40, 40–60, 60–80, and 80–100 cm) at 100 sampling points in one 15-year-old *Eucalyptus urophylla* hybrid plantation in southern China. A total of 1037 isolates of *Calonectria* present in all five soil layers were obtained from 93 of 100 sampling points. The 1037 isolates were identified based on DNA sequence comparisons of the translation elongation factor 1-alpha (*tef1*), β-tubulin (*tub2*), calmodulin (*cmdA*), and histone H3 (*his3*) gene regions, as well as the combination of morphological characteristics. These isolates were identified as *C. hongkongensis* (665 isolates; 64.1%), *C. aconidialis* (250 isolates; 24.1%), *C. kyotensis* (58 isolates; 5.6%), *C. ilicicola* (47 isolates; 4.5%), *C. chinensis* (2 isolates; 0.2%), and *C. orientalis* (15 isolates; 1.5%). With the exception of *C. orientalis*, which resides in the *C. brassicae* species complex, the other five species belonged to the *C. kyotensis* species complex. The results showed that the number of sampling points that yielded *Calonectria* and the number (and percentage) of *Calonectria* isolates obtained decreased with increasing depth of the soil. More than 84% of the isolates were obtained from the 0–20 and 20–40 cm soil layers. The deeper soil layers had comparatively lower numbers but still harbored a considerable number of *Calonectria*. The diversity of five species in the *C. kyotensis* species complex decreased with increasing soil depth. The genotypes of isolates in each *Calonectria* species were determined by *tef1* and *tub2* gene sequences. For each species in the *C. kyotensis* species complex, in most cases, the number of genotypes decreased with increasing soil depth. The 0–20 cm soil layer contained all of the genotypes of each species. To our knowledge, this study presents the first report of *C. orientalis* isolated in China. This species was isolated from the 40–60 and 60–80 cm soil layers at only one sampling point, and only one genotype was present. This study has enhanced our understanding of the species diversity and distribution characteristics of *Calonectria* in different soil layers.

## 1. Introduction

Species in the genus *Calonectria* (*Hypocreales*, *Nectriaceae*) are phytopathogenic fungi that cause serious losses to plant crops in tropical and subtropical regions of the world [1,2,3,4,5,6]. Many species of *Calonectria* are important pathogens of agricultural, horticultural, and forestry crops and these species occur in approximately 335 plant species in nearly 100 plant families [1]. Species of *Calonectria* have been isolated from soils, fruits, leaves, stems, and roots [1,4,7,8,9,10,11,12,13,14]. The fungi are best known as foliar, shoot, and root pathogens [1,2,4,5], and they are commonly associated with disease symptoms, including seedling damping-off, seedling rot, cutting rot, leaf spots, leaf blight, shoot blight, crown cankers, stem lesions, collar and root rots, and tuber rot [1,14,15,16,17,18,19,20,21,22,23].

Some species of *Calonectria* isolated from soils are important plant pathogens. *Calonectria ilicicola* is a soil-borne fungal pathogen of worldwide importance that causes black rot disease in peanut and red crown rot in soybean [21,24,25,26,27,28]. Recently, we isolated five *Calonectria* species, namely *C. aconidialis*, *C. auriculiformis*, *C. hongkongensis*, *C. pseudoreteaudii*, and *C. reteaudii*, from soils in a plantation of *Eucalyptus* trees [14]. Inoculation results showed that all five species caused leaf spot, leaf blight, and seedling rot to the tested *Eucalyptus* genotypes within three days [14].

Previous research results indicated a high level of species diversity of *Calonectria* in southern China, especially in soils [9,11,13,14,23]. Currently, a total of 125 *Calonectria* species have been described using DNA sequence-based phylogenetic analyses and morphological comparisons [5,13,29,30,31,32,33,34,35]. A total of 25 species of *Calonectria* have been identified and described in China based on DNA sequence data [5,9,11,13,14,36]. Of these species, 17 have been isolated from soils, with 11 from soils under plantation *Eucalyptus* trees [5,9,11,14].

Some *Calonectria* species can survive in soil for long periods, and microsclerotia are the primary survival structures [37]. Microsclerotia of some *Calonectria* species can survive in the absence of hosts for 15 years or more [38,39]. *Calonectria* microsclerotia have been recorded at depths of up to 66 cm below the soil surface [40]. Long-term survival and deep soil presence of microsclerotia are serious threats to the management of diseases caused by *Calonectria* species.

Understanding the diversity and distribution characteristics of *Calonectria* species in different soil layers will help us to clarify their potential long-term harm to plants and potential dissemination patterns. Very little research has been conducted concerning the distribution characteristics of microsclerotia in soils, and the few published studies have focused only on the surface soil [38,41]. In the past several years, studies have been conducted to understand *Calonectria* species diversity in forest soils [9,10,11,13,14,36], but all of the soil samples obtained for *Calonectria* isolation were collected from the 0–20 cm soil layer. In this study, a relatively large number of soil samples were collected from five different soil layers up to 100 cm depth in one 15-year-old *Eucalyptus urophylla* hybrid plantation. Isolates of *Calonectria* from this plantation were obtained and identified. The aims of this study were as follows: (1) to understand the species diversity of *Calonectria* in different soil layers; and (2) to understand the distribution characteristics of each *Calonectria* species in different soil layers.

## 2. Materials and Methods

### 2.1. Study Site, Soil Sampling, and Calonectria Isolation

This study was performed in a *Eucalyptus urophylla* hybrid plantation (21°15′31.74″ N, 110°06′35″ E; altitude 90 m) located in the South China Experimental Nursery, China Eucalypt Research Centre (CERC), ZhanJiang, GuangDong Province, China. The *Eucalyptus* plantation is located on the northern edge of the tropics, with a maritime monsoon climate [42]. The average annual precipitation is 1777 mm, and the period from May to October accounts for 84.1% of the annual precipitation. The annual average temperature is 23.4 °C (http://en.weather.com.cn; accessed date: 10 August 2021). The soil type is Rhodi-Udic Ferralosols, according to the Chinese Soil Taxonomy Classification [42,43]. The area of the *Eucalyptus* plantation is about 6 ha (400 × 150 m), and the planting density of *Eucalyptus* trees is 3 × 2 m. The *Eucalyptus* trees were 15 years old.

One hundred points in the *Eucalyptus* plantation were selected for soil sampling. The 100 points were randomly distributed in the plantation, and the distance between adjacent sampling points was 10 m. Soil samples were collected from five layers at each sampling point: 0–20, 20–40, 40–60, 60–80, and 80–100 cm. Two soil samples were collected in each soil layer for each sampling point. In total, 1000 soil samples were collected from the 100 sampling points. Each of the soil samples was placed in a resealable plastic bag and transferred to the laboratory for *Calonectria* isolation. The soil samples were collected from July to August 2020.

For *Calonectria* isolation, the collected soil was transferred into a plastic cylinder sampling cup (diameter = 4.5 cm, height = 5 cm, and volume = 80 mL) (Chengdu Rich Science Industry Co., Ltd., Chengdu, China); the soil sample occupied two-thirds of the volume of the whole sampling cup volume. The soil sample was moistened by spraying with sterile water and stirred evenly with a sterilized bamboo stick. *Medicago sativa* (alfalfa) seeds were scattered onto the soil surface after it was surface-disinfested (30 s in 75% ethanol and washed several times with sterile water) in the sampling cup. The sampling cup with soil and alfalfa seeds was incubated at 25 °C under 12 h of daylight and 12 h of darkness. After one week, sporulating conidiophores with typical morphological characteristics of *Calonectria* species [1] were produced on infected alfalfa tissue. Using a dissection microscope (AxioCam Stemi 2000C, Carl Zeiss, Germany), the single conidial mass was scattered onto 2% malt extract agar (MEA) (20 g malt extract powder and 20 g agar powder per liter of water: malt extract powder was obtained from Beijing Shuangxuan microbial culture medium products factory, Beijing, China; the agar powder was obtained from Beijing Solarbio Science & Technology Co., Ltd., Beijing, China) using a sterile needle. After incubation at 25 °C for three to four hours, the germinated conidia were individually transferred onto fresh MEA under the dissection microscope and incubated at 25 °C for one week to obtain single-conidium cultures. For each soil sample, the soil was transferred into two plastic sampling cups for *Calonectria* isolation.

### 2.2. DNA Extraction, PCR Amplification, and Sequencing

All isolates obtained in this study were used for DNA extraction and sequence comparisons. DNA was extracted from 10-day-old cultures. Mycelia were collected using a sterilized scalpel and transferred to 2-mL Eppendorf tubes. The total genomic DNA was extracted using the CTAB protocol described by van Burik and co-authors [44]. The extracted DNA was dissolved in 30 µL TE buffer (1 M Tris-HCl and 0.5 M EDTA, pH 8.0), and 2.5 µL RNase (10 mg/mL) was added at 37 °C for one hour to degrade RNA. Finally, the DNA concentration was measured using a NanoDrop 2000 spectrometer (Thermo Fisher Scientific, Waltham, MA, USA).

According to previous research results, sequences of partial gene regions of translation elongation factor 1-alpha (*tef1*) and β-tubulin (*tub2*), as well as calmodulin (*cmdA*) and histone H3 (*his3*), were used to successfully identify *Calonectria* species [5,14]. These four partial gene regions were amplified using the primer pairs EF1-728F/EF2, T1/CYLTUB1R, CAL-228F/CAL-2Rd, and CYLH3F/CYLH3R, respectively. The PCR procedure was conducted as described by Liu and Chen [36] and Wang and Chen [23].

To obtain accurate sequences for each of the sequenced isolates, all of the PCR products were sequenced in both forward and reverse directions using the same primers used for PCR amplification by the Beijing Genomics Institute, Guangzhou, China. All of the sequences obtained in this study were edited using MEGA v. 7.0 software [45] and were deposited in GenBank (https://www.ncbi.nlm.nih.gov; accessed date: 18 September 2021). The *tef1* and *tub2* gene regions were sequenced for all *Calonectria* isolates. The isolates were genotyped by the *tef1* and *tub2* sequences. Based on the genotypes generated by *tef1* and *tub2* sequences, up to eight isolates for each *tef1*-*tub2* genotype were selected for sequencing the *cmdA* and *his3* gene regions.

### 2.3. Multi-Gene Phylogenetic Analyses, Morphology, and Species Identification

A standard nucleotide BLAST search was conducted using the *tef1*, *tub2*, *cmdA*, and *his3* sequences to preliminarily identify the species from which the isolates were obtained in this study. Sequences of *tef1*, *tub2*, *cmdA*, and *his3* gene regions obtained in this study were compared with sequences of type specimen strains of published *Calonectria* species. Sequences of all of the published species in the relevant species complexes were used for sequence comparisons and phylogenetic analyses. The datasets of Liu and co-authors [5] were used as templates for analyses, while sequences of other recently described *Calonectria* species [13,32,33,34,35] were also used for sequence comparisons.

Sequences of each of the *tef1*, *tub2*, *cmdA*, and *his3* gene regions, as well as the combination of these four gene regions, were aligned using the online version of MAFFT v. 7 (http://mafft.cbrc.jp/alignment/server; accessed date: 7 August 2021) with the alignment strategy FFT-NS-i (Slow; interactive refinement method). Sequence alignments were manually edited using MEGA v. 7.0 software [45] after initial alignments.

For *Calonectria* species, maximum parsimony (MP) and maximum likelihood (ML) are frequently used for phylogenetic analyses [5,9,12,14]. Both MP and ML were used for phylogenetic analyses of sequence datasets of each of the four genes and the combination of the four gene regions in order to test whether the analysis results between the two methods were consistent. The MP and ML analyses were conducted by the methods described by Liu and Chen [36]. Phylogenetic trees were viewed by MEGA v. 7.0 [45]. Sequence data of two isolates of *Curvicladiella cignea* (CBS 109167 and CBS 109168) were used as outgroups [5].

The isolates selected for sequencing *tef1*, *tub2*, *cmdA*, and *his3* gene regions were used for morphological studies. Size of macroconidia and width of vesicles are the most typical asexual characteristics used for morphological comparisons for species of *Calonectria* [5,9,11,13,14,29,36]. In order to induce asexual structures, isolates were cultured on 2% MEA in Petri dishes at 25 °C for 10 days. Sterile water was then added to the Petri dishes, and a sterilized, soft-bristled paintbrush was used to dislodge the mycelium from the agar surface. The water was then removed, and the dishes were placed upside down and incubated at 25 °C for 2–3 days. This resulted in asexual structures being produced on the surface of the cultures for some *Calonectria* isolates, a pattern that has been noted for *Calonectria pteridis* by Graça and co-authors [46] and for *Calonectria pentaseptata* (synonymized as a synonym of *C. pseudoreteaudii* in Liu and co-authors [5]) by Wang and Chen [23]. Fifty measurements of macroconidia and vesicles were measured for the selected isolates that produced abundant macroconidia and vesicles.

### 2.4. Calonectria Species Diversity in Different Soil Layers

After all of the *Calonectria* isolates were identified, the number of isolates present in each identified species was counted. The species diversity associated with soil layers was computed. The distribution characteristics of each *Calonectria* species in each soil layer were recorded, including the number of sampling points from which each *Calonectria* species was obtained and the number of isolates of each *Calonectria* species in each of the five soil layers.

### 2.5. Genotyping of Isolates within Each Calonectria Species

After all of the *Calonectria* isolates were identified, we examined the genotype diversity of each identified *Calonectria* species in the five different soil layers. The genotypes of isolates within each species were determined based on *tef1* and *tub2* sequences, and the number of isolates belonging to each genotype was recorded.

### 2.6. Genotype Diversity of Calonectria Species in Different Soil Layers

Based on the results of genotype analysis of each isolate determined by the sequences of *tef1* and *tub2* gene regions, the numbers of genotypes of each *Calonectria* species in different soil layers were counted. To investigate possible evolutionary relationships among the observed *tef1*–*tub2* genotypes for the *Calonectria* species identified in this study with the most dominant species, minimum spanning networks (MSN) were constructed using Bruvo’s distance with the R packages poppr and ape [47,48].

## 3. Results

### 3.1. Soil Sampling and Calonectria Isolation

One thousand soil samples from 100 sample points were collected from the *E. urophylla* hybrid plantation, with 200 soil samples from each of the five soil layers. For each soil sample, two plastic sampling cups with soil and alfalfa seeds were used for the incubation of *Calonectria*. After the conidia were transferred onto fresh MEA and incubated at 25 °C, more than 90% of the conidia germinated within four hours. For each sampling cup, one to four single conidia were transferred onto fresh MEA to obtain one to four single-conidium cultures. In total, *Calonectria* fungi were isolated from 93 sampling points in the plantation; the totals were 92, 40, 20, 7, and 5 from the 0–20, 20–40, 40–60, 60–80, and 80–100 cm soil layers, respectively (Appendix A, Appendix A). One thousand and thirty-seven isolates of *Calonectria* were obtained, with 564 (54.4%), 310 (29.9%), 107 (10.3%), 28 (2.7%), and 28 isolates (2.7%) from the 0–20, 20–40, 40–60, 60–80, and 80–100 cm soil layers, respectively, and 84.3% of the isolates were distributed in the 0–20 and 20–40 cm soil layers (Table 1, Appendix A, Figure 1). From the results, it was clear that the number of sampling points that yielded *Calonectria* and the number (and percentage) of *Calonectria* isolates obtained decreased with increasing soil depth (Appendix A, Figure 1).

### 3.2. Sequencing

The *tef1* and *tub2* genes were amplified for all the 1037 isolates obtained in this study (Appendix A). Twenty-two genotypes were generated based on *tef1* and *tub2* gene sequences (Table 2). Depending on the isolate number of each *tef1*-*tub2* genotype, one to eight isolates of each genotype were selected; finally, 85 isolates in total were selected to sequence the *cmdA* and *his3* gene regions (Table 3). The sequence fragments were approximately 500, 565, 685, and 440 bp for the *tef1*, *tub2*, *cmdA*, and *his3* gene regions, respectively.

### 3.3. Multi-Gene Phylogenetic Analyses, Morphology, and Species Identification

The standard nucleotide BLAST search results conducted using the *tef1*, *tub2*, *cmdA*, and *his3* sequences showed that the isolates obtained in the current study belonged to two species complexes of *Calonectria*, namely, the *C. kyotensis* species complex and the *C. brassicae* species complex. The 85 *Calonectria* isolates with four gene regions sequenced were used for phylogenetic analyses (Table 3). Based on the recently published results in Liu and co-authors [5] and Crous and co-authors [34], sequences of *tef1*, *tub2*, *cmdA*, and *his3* of published species in the *C. kyotensis* species complex and *C. brassicae* species complex, respectively, were used for sequence comparisons and phylogenetic analyses (Table 4).

The partition homogeneity test (PHT) comparing the *tef1*, *tub2*, *cmdA*, and *his3* gene combination datasets generated a *p*-value of 0.001, indicating that the accuracy of the combined datasets did not suffer relative to the individual partitions [60]. Thus, sequences of the four loci were combined for analyses. Between the MP and ML trees, the overall topologies were similar for the phylogenetic trees based on *tef1*, *tub2*, *cmdA*, and *his3* individually and the combination datasets, but the relative positions of some *Calonectria* species slightly differed. The five ML trees are presented in Figure 2 and Appendix A. The numbers of taxa and parsimony-informative characters, statistical values of the MP analyses, and parameters of the best-fit substitution models of ML analyses are provided in Table 5.

The phylogenetic analyses showed that the 85 *Calonectria* isolates were clustered in six groups (Group A, Group B, Group C, Group D, Group E, and Group F) based on *tef1*, *tub2*, *cmdA*, *his3,* and combined *tef1*/*tub2*/*cmdA*/*his3* analyses (Figure 2, Appendix A). The analyses showed that isolates in Groups A, B, C, D, and E belonged to the *C. kyotensis* species complex. Isolates in Groups A, B, C, and E were clustered with or were closest to *C. hongkongensis*, *C. kyotensis*, *C. chinensis,* and *C. ilicicola*, respectively, based on the *tef1*, *tub2*, *cmdA*, *his3,* and combined *tef1*/*tub2*/*cmdA*/*his3* trees (Figure 2, Appendix A). Therefore, isolates in Groups A, B, C, and E were identified as *C. hongkongensis*, *C. kyotensis*, *C. chinensis,* and *C. ilicicola*, respectively. Isolates in Group D were clustered in two sub-groups, sub-group D1 and sub-group D2, in the *tub2* tree. Isolates in sub-group D1 were clustered with or were closest to *C. aconidialis*; isolates in sub-group D2 were clustered with *C. asiatica* (Appendix A); isolates in Group D were clustered with or were closest to *C. aconidialis* based on the *tef1*, *cmdA*, *his3,* and combined *tef1*/*tub2*/*cmdA*/*his3* trees (Figure 2, Appendix A). Isolates in Group D were identified as *C. aconidialis*. Isolates in Group F belonged to the *C. brassicae* species complex. These isolates were consistently only clustered with *C. orientalis* based on the *tef1*, *tub2*, *his3,* and combined *tef1*/*tub2*/*cmdA*/*his3* trees and were clustered with both *C. orientalis* and *C. brassicae* in the *cmdA* tree (Figure 2, Appendix A). Isolates in Group F were identified as *C. orientalis*.

Based on the results of phylogenetic analyses and induction of asexual structures, 17 isolates representing six *Calonectria* species were selected for macroconidia and vesicle morphological comparisons (Table 3 and Table 6). These representative isolates could be classified into two groups based on the shape of the vesicles. Isolates of *C. aconidialis*, *C. chinensis*, *C. hongkongensis*, *C. ilicicola*, and *C. kyotensis* produce sphaeropedunculate vesicles, while the vesicles of *C. orientalis* are typically clavate. With the exception of *C. ilicicola* isolates, which produce 1(–3) septate macroconidia, isolates of the other five species all produced one septate macroconidium (Table 6). The shape of the vesicle and the number of macroconidia septations for each of the six *Calonectria* species found in this study were consistent with the described strains of relevant species in previous studies [1,9,29,49] (Table 6).

The morphological comparisons showed that significant variation existed in the size of macroconidia or width of vesicles among some isolates of each species of *C. aconidialis*, *C. hongkongensis*, and *C. kyotensis* identified in this study. For example, the macroconidia of *C. aconidialis* isolate CSF20985 were much longer than those of the other two tested *C. aconidialis* isolates CSF20323 and CSF20376. In *C. hongkongensis*, the macroconidia of isolate CSF20383 were longer than those of the other four isolates; the vesicles of isolate CSF20924 were wider than those of other isolates. In *C. kyotensis*, the macroconidia of isolate CSF20276 were much longer than those of isolate CSF21191 (Table 6).

The measurement results further showed that macroconidia size and vesicle width of isolates of some species obtained in this study were not always similar to the originally described strains of the same *Calonectria* species. For example, the macroconidia lengths of isolates of *C. chinensis* and *C. orientalis* obtained in this study were much shorter than the originally described strains of the relevant species [29,49] (Table 6).

### 3.4. Calonectria Species Diversity in Different Soil Layers

Based on the sequence comparisons of *tef1*, *tub2*, *cmdA,* and *his3* sequences, the 1037 *Calonectria* isolates were identified as *C. hongkongensis* (665 isolates; 64.1%), *C. aconidialis* (250 isolates; 24.1%), *C. kyotensis* (58 isolates; 5.6%), *C. ilicicola* (47 isolates; 4.5%), *C. chinensis* (2 isolates; 0.2%), and *C. orientalis* (15 isolates; 1.5%) (Table 1). *Calonectria hongkongensis* was dominant, followed by *C. aconidialis.* Each of the two dominant species was isolated from more than or close to 50% of all of the sampling points, and the two species accounted for 88.2% of all of the *Calonectria* isolates obtained in this study (Table 1, Appendix A, Figure 3). Both *C. chinensis* and *C. orientalis* were only isolated from one sampling point; *C. chinensis* was only isolated from the 0–20 cm soil layer, and only two isolates were obtained; *C. orientalis* was isolated from the soil layers of 40–60 and 60–80 cm, and 11 and 4 isolates in the two soil layers were obtained, respectively (Table 1, Appendix A, Figure 3).

With the exception of *C. orientalis* in the *C. brassicae* species complex, the diversity of species in the *C. kyotensis* species complex decreased with increasing soil depth. Five, four, four, four, and two species were identified in the soil layers of 0–20, 20–40, 40–60, 60–80, and 80–100 cm, respectively (Table 1, Appendix A).

For each of the five species in the *C. kyotensis* species complex, the number of sampling points containing *Calonectria* decreased with increasing depth of the soil, with the exception of *C. hongkongensis* in soil layers of 60–80 cm (2 sampling points) and 80–100 cm (4 sampling points) (Appendix A, Figure 4A); the number of isolates obtained decreased with increasing soil depth, with the exception of *C. hongkongensis* in the 60–80 cm soil layer (8 isolates) and 80–100 cm (20 isolates) as well as *C. ilicicola* in the 0–20 cm (16 isolates) and 20–40 cm (19 isolates) soil layers (Table 1, Figure 4B). Most isolates were obtained from the soil layers 0–20 and 20–40 cm, accounting for 86.6%, 85.6%, 81%, 74.5%, and 100% of all of the obtained isolates within each species of *C. hongkongensis*, *C. aconidialis*, *C. kyotensis*, *C. ilicicola*, and *C. chinensis*, respectively (Figure 5).

### 3.5. Genotyping of Isolates within Each Calonectria Species

For the 1037 *Calonectria* isolates obtained and identified in this study, the genotype results based on *tef1* and *tub2* sequences indicated that 11, 3, 3, 3, 1, and 1 genotype(s) existed in *C. hongkongensis*, *C. aconidialis*, *C. kyotensis*, *C. ilicicola*, *C. chinensis*, and *C. orientalis*, respectively (Table 2). The isolates presenting the dominant genotype (genotype AA) accounted for 84.4%, 62.4%, 56.9%, 55.3%, 100%, and 100% of all of the isolates obtained from *C. hongkongensis*, *C. aconidialis*, *C. kyotensis*, *C. ilicicola*, *C. chinensis*, and *C. orientalis*, respectively (Table 2).

### 3.6. Genotype Diversity of Calonectria Species in Different Soil Layers

The *tef1*-*tub2* genotypes of each *Calonectria* species in each soil layer are listed in Table 7 and are shown in Figure 6. For each species in the *C. kyotensis* species complex, the results showed that the number of genotypes decreased with increasing soil depth, with the exception of *C. hongkongensis* and *C. aconidialis* in the 60–80 cm (one genotype) and 80–100 cm (two genotypes) soil layers (Table 7, Figure 6A,B); the 0–20 cm soil layer contained all of the genotypes of each species in the *C. kyotensis* complex (Table 7, Figure 6A–E). For the genotype with the most isolates of each species in the *C. kyotensis* complex, the majority of isolates were obtained from 0–20 cm soil layer, with the exception of *C. ilicicolla* (Table 7, Figure 6A–E). Only one genotype of *C. orientalis* was present in the 40–60 and 60–80 cm soil layers (Table 7, Figure 6F).

The minimum spanning network (MSN) analysis was conducted for *C. hongkongensis*, which was considered as the dominant species identified in this study. The analysis revealed that most isolates of *C. hongkongensis* were genotype AA (561 isolates), followed by genotypes DA (31 isolates) and AF (20 isolates); genotype AA was present in the isolates from all five soil layers; genotypes AB, AC, AE, AH, and BA were present only in the isolates from the 0–20 cm soil layer, and the other genotypes were present in isolates from two to four soil layers. Isolates from the 0–20 cm soil layer contained all of the genotypes (Figure 7).

## 4. Discussion

In this study, more than 1000 *Calonectria* isolates were obtained from five soil layers at 100 sampling points in one *Eucalyptus* plantation. All of the isolates were identified based on DNA sequence comparisons of multiple gene regions. Six *Calonectria* species were identified, namely, *C. aconidialis*, *C. chinensis*, *C. hongkongensis*, *C. ilicicola,* and *C. kyotensis* in the *C. kyotensis* species complex, and *C. orientalis* in the *C. brassicae* species complex. *Calonectria hongkongensis* (64.1% of all of the isolates) was the dominant species, followed by *C. aconidialis* (24.1% of all of the isolates). To our knowledge, this is the first report of *C. orientalis* in China. The species diversity and distribution characteristics of the six species in different soil layers were clarified. The results showed that the number of sampling points from which *Calonectria* was obtained, and the number of *Calonectria* isolates obtained decreased with increasing depth of the soil. The majority of isolates (84.3% of all the isolates) were obtained from soil layers of 0–20 and 20–40 cm. The diversity of the five species in the *C. kyotensis* species complex decreased with increasing soil depth. For each species in the *C. kyotensis* species complex, in most cases, the number of genotypes decreased with increasing soil depth, and the 0–20 cm soil layer contained all of the genotypes of each species.

Five species, namely, *C. aconidialis*, *C. chinensis*, *C. hongkongensis*, *C. ilicicola*, and *C. kyotensis,* in the *C. kyotensis* species complex were isolated from the soil of the *Eucalyptus* plantation in this study. These five species have been frequently isolated from soils in several other regions in southern China, especially from soils in *Eucalyptus* plantations [9,11,14,49]. *Calonectria ilicicola* is considered as a soil-borne fungal pathogen that has been isolated from a number of diseased plant species in China [21,61]. This study presents the first record of *C. ilicicola* isolated from soil in a *Eucalyptus* plantation. Results of this and previous studies suggest that all five of the species in the *C. kyotensis* species complex are potentially widely distributed in *Eucalyptus* plantation soils in other regions of southern China [9,11,14].

This study is the first report of *C. orientalis* in China, and this species is the first *Calonectria* species in the *C. brassicae* species complex found in China. *Calonectria orientalis* has been isolated from soil in Indonesia [29]. Some other species in the *C. brassicae* species complex have also been frequently isolated from soils. With the exception of *C. orientalis*, the other species in the *C. brassicae* species complex isolated from soils were all from Ecuador and Brazil in South America [5,10,29,30,31,56]. Most of the *Calonectria* species in the *C. brassicae* species complex have only been isolated from South America but not from Asia [5] and *C. orientalis*, in this study, was only isolated from one of the 100 sampling points. These results suggest that *C. orientalis* is not widely distributed in China.

For the five species in the *C. kyotensis* species complex, the results of this study indicate that the diversity of the five species decreased with increasing soil depth, and the number of sampling points containing *Calonectria* and the number of *Calonectria* isolates obtained also decreased with soil depth. Most isolates were obtained from the 0–20 and 20–40 cm soil layers. In most cases, the number of genotypes decreased with increasing soil depth for each species, and the 0–20 cm soil layer contained all of the genotypes of each species. These results suggest that 0–20 cm is the best soil depth for *Calonectria* isolation and for examining the species and genotype diversity of *Calonectria* in soils in *Eucalyptus* plantations in southern China. In several previous studies specialized in the research on *Calonectria* species diversity, soil samples were also exclusively collected from the surface layer, all from the 0–20 cm layer [9,10,11,13,14,36]. These studies have characterized the diversity of *Calonectria* species well. Results of a number of other studies indicated that microbial diversity and richness are typically affected by the soil depth [62,63,64,65,66,67], and shallower layers usually have a higher level of microbial diversity [62,63,66,67,68]. This pattern is consistent with the results of the present study. A possible reason for the vertical distribution of soil microbes is the harsher environment in deeper soil layers, where the soil density is higher, oxygen concentrations are lower, and carbon and nutrients are less available [69]. For *Calonectria*, which includes some important pathogens of various agricultural, horticultural, and forestry crops worldwide, as well as for other genera of fungi in forests, no systematic research has been conducted to examine the species diversity and distribution characteristics in different soil layers. This study showed that the deeper soil layers had comparatively fewer but still contained many *Calonectria*. It remains unknown whether the *Calonectria* were originally distributed in deeper soil layers or whether the fungi in deeper soil layers migrated from surface layers, perhaps through the infiltration of rainwater. Studies on the population diversity differences among different soil layers should be conducted to address this question. Furthermore, the *Calonectria* distributed in deeper soil layers increase the challenge of controlling the diseases caused by these fungi.

This study examined the species diversity and distribution characteristics of *Calonectria* in five soil layers in a *Eucalyptus* plantation in southern China. Six species were isolated from soils in a relatively small *Eucalyptus* plantation, indicating that the diversity of *Calonectria* species in these soils in southern China is relatively high. This study also revealed that the species diversity and number of genotypes of each *Calonectria* species decreased with increasing soil depth, a pattern that helps us to understand the distribution characteristics of *Calonectria* species in different layers of soil. For some *Calonectria* species, there were relatively large numbers of isolates obtained from different soil layers, especially for *C. hongkongensis* and *C. aconidialis* in the 0–20, 20–40, and 40–60 cm soil layers. The genetic structures and population biology of these species in the different soil layers are unknown, but additional studies may increase our understanding of the distribution characteristics and dissemination patterns of *Calonectria* species.

## Figures and Tables

**Figure 1 jof-07-00857-f001:**
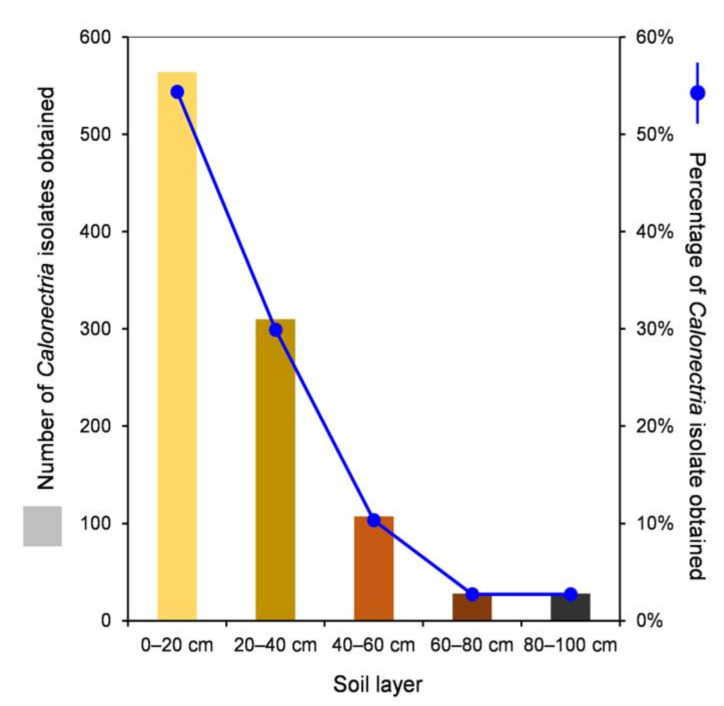
Numbers and percentages of *Calonectria* isolates obtained in each of the five soil layers.

**Figure 2 jof-07-00857-f002:**
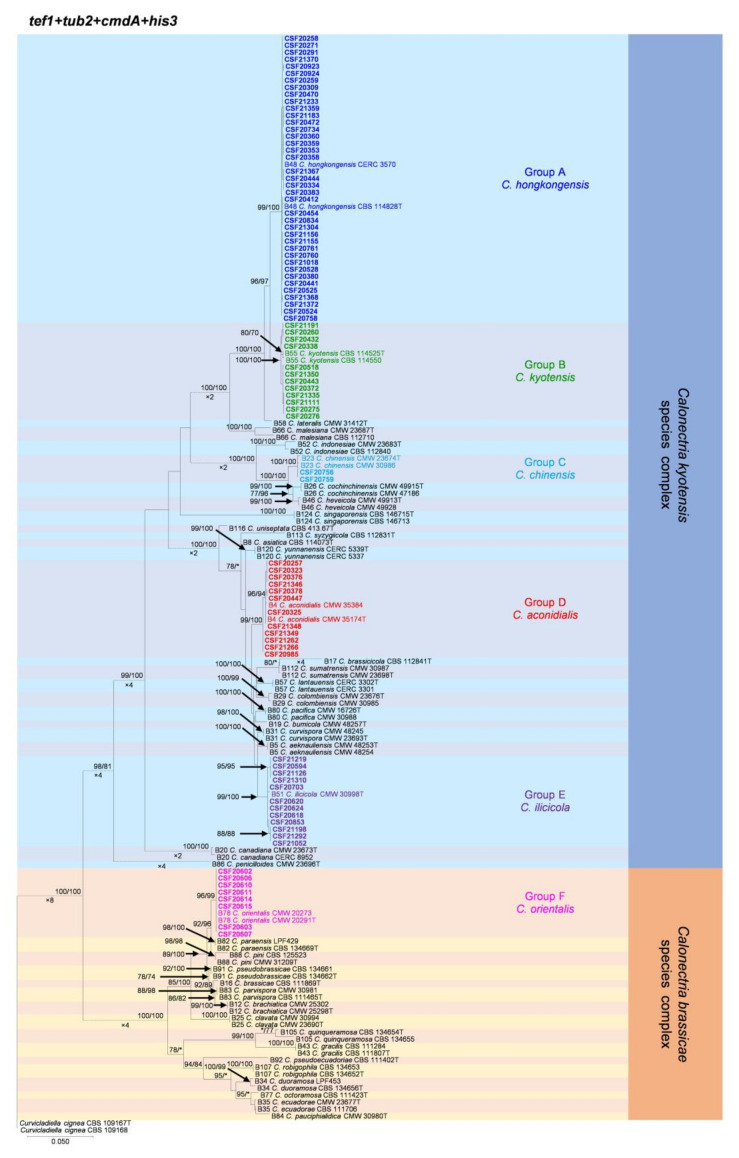
Phylogenetic tree of *Calonectria* species based on maximum likelihood (ML) analyses of the dataset of combined *tef1*, *tub2*, *cmdA*, and *his3* gene sequences in this study. Bootstrap support values ≥ 70% are presented above the branches as follows: ML/MP. Bootstrap values < 70% or absent are marked with “*”. Isolates highlighted in six different colors, and bold were obtained in this study. Ex-type isolates are marked with “T”. The “B” species codes are consistent with the recently published results in Liu and co-authors [5]. *Curvicladiella cignea* (CBS 109167 and CBS 109168) was used as the outgroup taxon.

**Figure 3 jof-07-00857-f003:**
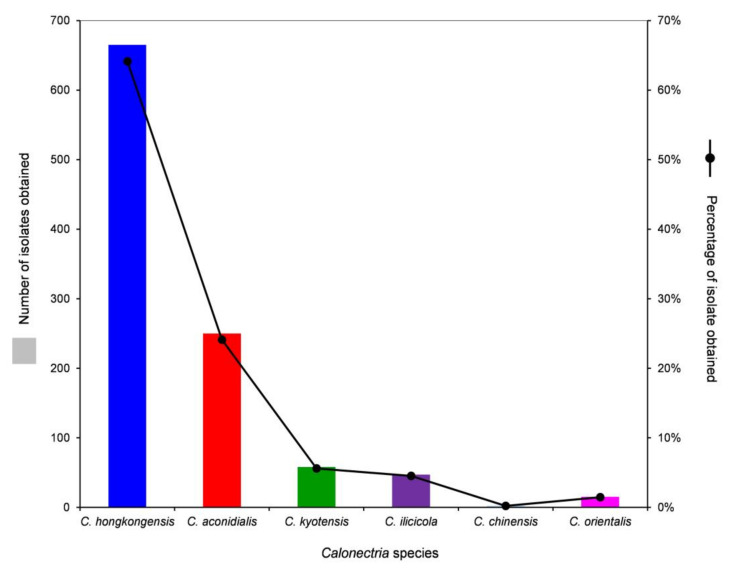
Numbers and percentages of isolates obtained for each *Calonectria* species from all soil samples collected.

**Figure 4 jof-07-00857-f004:**
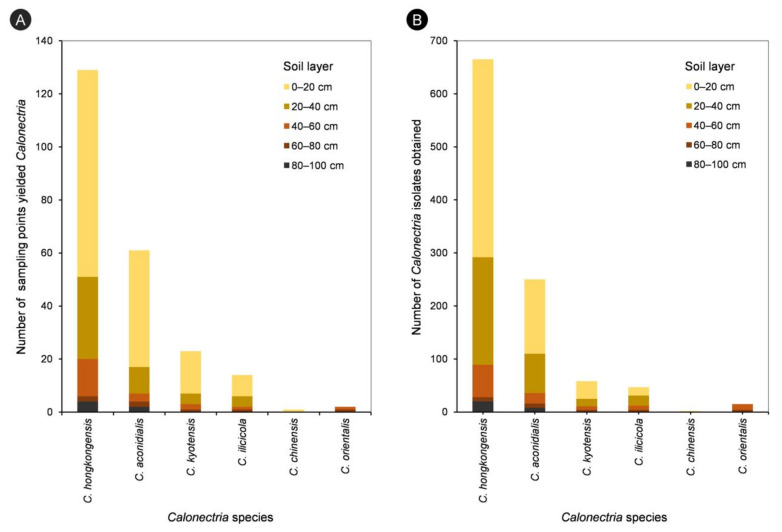
Number of sampling points yielded different *Calonectria* species in each of the five soil layers (**A**), and numbers of isolates obtained for different *Calonectria* species in each of the five soil layers (**B**).

**Figure 5 jof-07-00857-f005:**
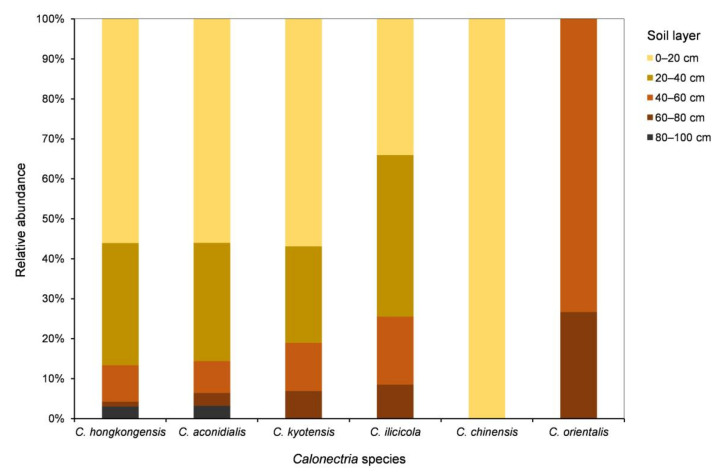
Relative abundances of each *Calonectria* species in each of the five soil layers. Relative abundance was based on the proportional frequencies of isolates of each *Calonectria* species in each soil layer.

**Figure 6 jof-07-00857-f006:**
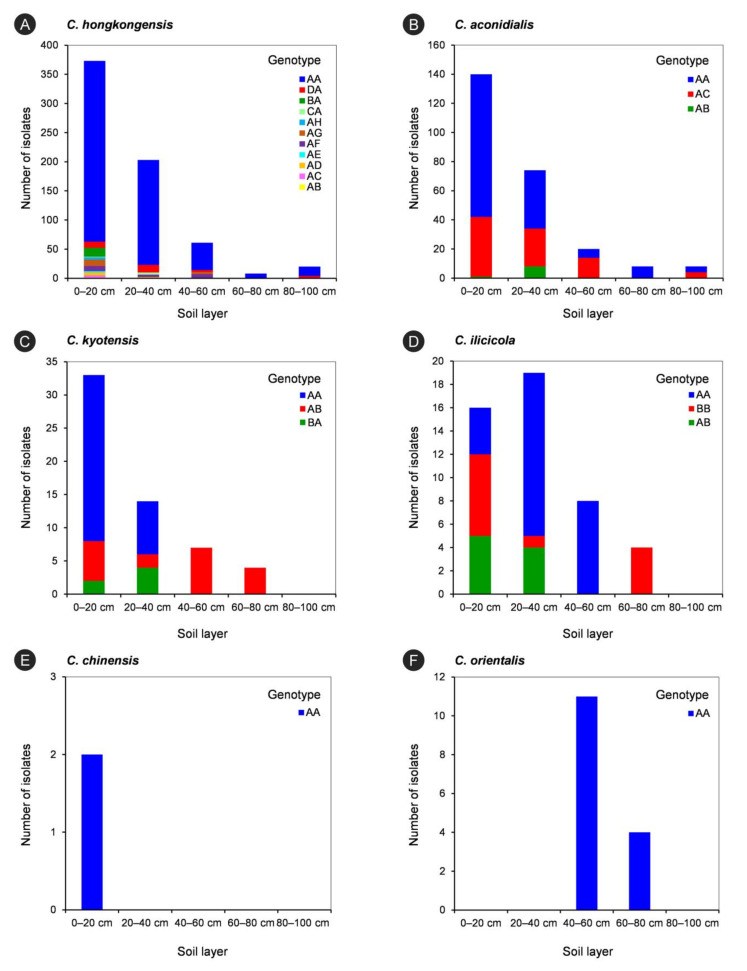
The isolate numbers of each genotype of each *Calonectria* species in five soils layers. The genotypes were determined by DNA sequences of *tef1* and *tub2* gene regions. (**A**): *C. hongkongensis*; (**B**): *C. aconidialis*; (**C**): *C. kyotensis*; (**D**) *C. ilicicola*; (**E**): *C. chinensis*; (**F**): *C. orientalis*.

**Figure 7 jof-07-00857-f007:**
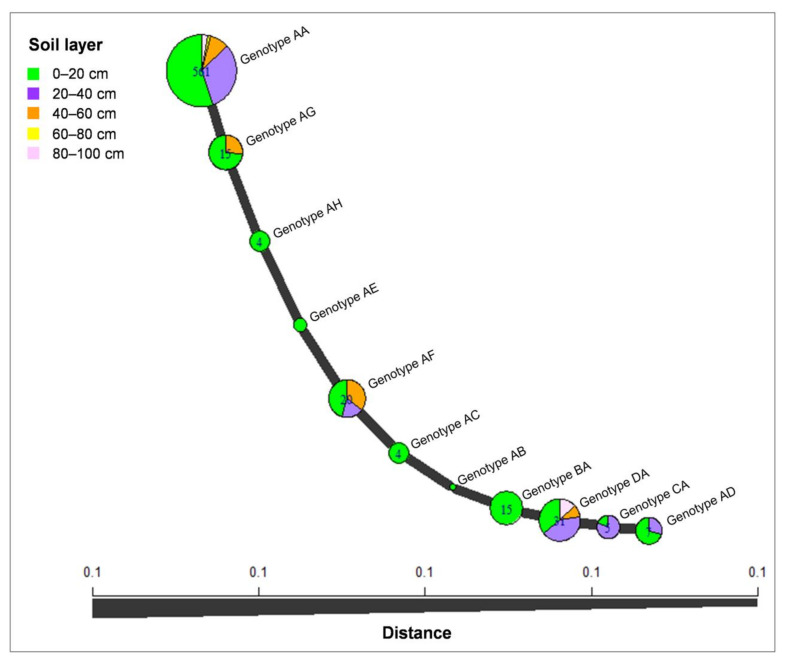
Minimum spanning network constructed using Bruvo’s distances showing that the *C. hongkongensis* isolates were grouped into 11 genotypes based on *tef1* and *tub2* sequences. The size of a node is proportional to the number of represented *tef1*-*tub2* genotypes.

**Table 1 jof-07-00857-t001:** Number of isolates obtained for each *Calonectria* species from each soil layer.

Soil Layer	*C. hongkongensis*	*C. aconidialis*	*C. kyotensis*	*C. ilicicola*	*C. chinensis*	*C. orientalis*	All six *Calonectria* species	Percentage
0–20 cm	373	140	33	16	2	0	564	54.4%
20–40 cm	203	74	14	19	0	0	310	29.9%
40–60 cm	61	20	7	8	0	11	107	10.3%
60–80 cm	8	8	4	4	0	4	28	2.7%
80–100 cm	20	8	0	0	0	0	28	2.7%
All five soil layers	665	250	58	47	2	15	1037	
Percentage	64.1%	24.1%	5.6%	4.5%	0.2%	1.5%		

**Table 2 jof-07-00857-t002:** Isolate numbers of each genotype from each *Calonectria* species.

*Calonectria* Species	Number of Genotypes Determined by *tef1* and *tub2* Gene Sequences	Genotype Determined by *tef1* and *tub2* Gene Sequences	Number of Isolates of Each Genotype	Number of isolates of Each *Calonectria* Species
*C. hongkongensis*	11	AA	561	665
		AB	1	
		AC	4	
		AD	7	
		AE	2	
		AF	20	
		AG	15	
		AH	4	
		BA	15	
		CA	5	
		DA	31	
*C. aconidialis*	3	AA	156	250
		AB	9	
		AC	85	
*C. kyotensis*	3	AA	33	58
		AB	19	
		BA	6	
*C. ilicicola*	3	AA	26	47
		AB	9	
		BB	12	
*C. chinensis*	1	AA	2	2
*C. orientalis*	1	AA	15	15
All six *Calonectria* species	22		1037	1037

**Table 3 jof-07-00857-t003:** Isolates sequenced and used for phylogenetic analyses and morphological studies in this study.

Identity	Genotype ^1^	Isolate No. ^2^	Sampling Point No. ^3^	Soil Layer	Sample and Isolate Information ^4^	Collectors	GenBank Accession No. ^5^			
							*tef1*	*tub2*	*cmdA*	*his3*
*C. aconidialis*	AAAA	CSF20325	6	0–20 cm	20200711-1-(3)_0–20 cm_A_R2_SC2	S.F. Chen, L.L. Liu, J.L. Han, Y Liu, and X.Y. Liang	OK167700	OK168737	OK169148	OK169232
*C. aconidialis*	AAAA	CSF21348	98	0–20 cm	20200816-1-(6)_0–20 cm_A_R2_SC1	L.L. Liu, J.L. Han, and L.S. Sun	OK167855	OK168892	OK169151	OK169235
*C. aconidialis*	AACA	CSF20378	9	0–20 cm	20200711-1-(6)_0–20 cm_A_R2_SC1	S.F. Chen, L.L. Liu, J.L. Han, Y Liu, and X.Y. Liang	OK167701	OK168738	OK169149	OK169233
*C. aconidialis*	AACA	CSF20447	11	0–20 cm	20200715-1-(1)_0–20 cm_B_R2_SC2	S.F. Chen, L.L. Liu, J.L. Han, L.S. Sun, and W.W. Li	OK167704	OK168741	OK169150	OK169234
*C. aconidialis*	ABBA	CSF20985 ^6^	68	20–40 cm	20200811-1-(4)_0–40 cm_B_R1_SC3	L.L. Liu, J.L. Han, and L.S. Sun	OK167856	OK168893	OK169152	OK169236
*C. aconidialis*	ABBA	CSF21262	93	20–40 cm	20200816-1-(1)_0–40 cm_B_R1_SC1	L.L. Liu, J.L. Han, and L.S. Sun	OK167857	OK168894	OK169153	OK169237
*C. aconidialis*	ABBA	CSF21266	93	20–40 cm	20200816-1-(1)_0–40 cm_B_R2_SC2	L.L. Liu, J.L. Han, and L.S. Sun	OK167861	OK168898	OK169154	OK169238
*C. aconidialis*	ABBA	CSF21349	98	0–20 cm	20200816-1-(6)_0–20 cm_A_R2_SC2	L.L. Liu, J.L. Han, and L.S. Sun	OK167864	OK168901	OK169155	OK169239
*C. aconidialis*	ACAA	CSF20257	1	0–20 cm	20200709-1-(1)_0–20 cm_A_R1_SC1	S.F. Chen, L.L. Liu, J.L. Han, Y Liu, and X.Y. Liang	OK167865	OK168902	OK169156	OK169240
*C. aconidialis*	ACAA	CSF20323 ^6^	6	0–20 cm	20200711-1-(3)_0–20 cm_A_R1_SC1	S.F. Chen, L.L. Liu, J.L. Han, Y Liu, and X.Y. Liang	OK167866	OK168903	OK169157	OK169241
*C. aconidialis*	ACAA	CSF20376 ^6^	9	0–20 cm	20200711-1-(6)_0–20 cm_A_R1_SC1	S.F. Chen, L.L. Liu, J.L. Han, Y Liu, and X.Y. Liang	OK167868	OK168905	OK169158	OK169242
*C. aconidialis*	ACAA	CSF21346	98	0–20 cm	20200816-1-(6)_0–20 cm_A_R1_SC1	L.L. Liu, J.L. Han, and L.S. Sun	OK167946	OK168983	OK169159	OK169243
*C. chinensis*	AAAA	CSF20756 ^6^	52	0–20 cm	20200809-1-(2)_0–20 cm_A_R2_SC1	L.L. Liu, J.L. Han, and L.S. Sun	OK168055	OK169092	OK169184	OK169268
*C. chinensis*	AAAA	CSF20759 ^6^	52	0–20 cm	20200809-1-(2)_0–20 cm_A_R2_SC4	L.L. Liu, J.L. Han, and L.S. Sun	OK168056	OK169093	OK169185	OK169269
*C. hongkongensis*	AAAA	CSF20258	1	0–20 cm	20200709-1-(1)_0–20 cm_A_R1_SC2	S.F. Chen, L.L. Liu, J.L. Han, Y Liu, and X.Y. Liang	OK167035	OK168072	OK169109	OK169194
*C. hongkongensis*	AAAA	CSF20271	2	0–20 cm	20200709-1-(2)_0–20 cm_A_R1_SC1	S.F. Chen, L.L. Liu, J.L. Han, Y Liu, and X.Y. Liang	OK167044	OK168081	OK169110	OK169195
*C. hongkongensis*	AAAA	CSF20291	3	0–20 cm	20200709-1-(3)_0–20 cm_A_R2_SC1	S.F. Chen, L.L. Liu, J.L. Han, Y Liu, and X.Y. Liang	OK167056	OK168093	OK169111	OK169196
*C. hongkongensis*	AAAA	CSF21370	100	0–20 cm	20200816-1-(8)_0–20 cm_A_R2_SC2	L.L. Liu, J.L. Han, and L.S. Sun	OK167588	OK168625	OK169112	OK169197
*C. hongkongensis*	ABA-	CSF20758	52	0–20 cm	20200809-1-(2)_0–20 cm_A_R2_SC3	L.L. Liu, J.L. Han, and L.S. Sun	OK167596	OK168633	OK169113	– ^7^
*C. hongkongensis*	ACAA	CSF20524	17	0–20 cm	20200715-1-(7)_0–20 cm_B_R1_SC1	S.F. Chen, L.L. Liu, J.L. Han, L.S. Sun, and W.W. Li	OK167597	OK168634	OK169114	OK169198
*C. hongkongensis*	ACAA	CSF20525	17	0–20 cm	20200715-1-(7)_0–20 cm_B_R1_SC2	S.F. Chen, L.L. Liu, J.L. Han, L.S. Sun, and W.W. Li	OK167598	OK168635	OK169115	OK169199
*C. hongkongensis*	ACAB	CSF21368	100	0–20 cm	20200816-1-(8)_0–20 cm_A_R1_SC2	L.L. Liu, J.L. Han, and L.S. Sun	OK167599	OK168636	OK169116	OK169200
*C. hongkongensis*	ACAB	CSF21372	100	0–20 cm	20200816-1-(8)_0–20 cm_B_R1_SC2	L.L. Liu, J.L. Han, and L.S. Sun	OK167600	OK168637	OK169117	OK169201
*C. hongkongensis*	ADAA	CSF20412	10	0–20 cm	20200711-1-(7)_0–20 cm_B_R1_SC1	S.F. Chen, L.L. Liu, J.L. Han, Y Liu, and X.Y. Liang	OK167601	OK168638	OK169118	OK169202
*C. hongkongensis*	ADAA	CSF20454	11	20–40 cm	20200715-1-(1)_0–40 cm_A_R2_SC3	S.F. Chen, L.L. Liu, J.L. Han, L.S. Sun, and W.W. Li	OK167602	OK168639	OK169119	OK169203
*C. hongkongensis*	ADAA	CSF20834	60	0–20 cm	20200810-1-(4)_0–20 cm_B_R1_SC1	L.L. Liu, J.L. Han, and L.S. Sun	OK167604	OK168641	OK169120	OK169204
*C. hongkongensis*	ADAA	CSF21304	96	0–20 cm	20200816-1-(4)_0–20 cm_A_R2_SC1	L.L. Liu, J.L. Han, and L.S. Sun	OK167607	OK168644	OK169121	OK169205
*C. hongkongensis*	AEAA	CSF20923	65	0–20 cm	20200811-1-(1)_0–20 cm_A_R1_SC1	L.L. Liu, J.L. Han, and L.S. Sun	OK167608	OK168645	OK169122	OK169206
*C. hongkongensis*	AEAA	CSF20924 ^6^	65	0–20 cm	20200811-1-(1)_0–20 cm_A_R1_SC2	L.L. Liu, J.L. Han, and L.S. Sun	OK167609	OK168646	OK169123	OK169207
*C. hongkongensis*	AFAA	CSF20259	1	0–20 cm	20200709-1-(1)_0–20 cm_A_R2_SC1	S.F. Chen, L.L. Liu, J.L. Han, Y Liu, and X.Y. Liang	OK167610	OK168647	OK169124	OK169208
*C. hongkongensis*	AFAA	CSF20309	4	0–20 cm	20200711-1-(1)_0–20 cm_A_R1_SC1	S.F. Chen, L.L. Liu, J.L. Han, Y Liu, and X.Y. Liang	OK167611	OK168648	OK169125	OK169209
*C. hongkongensis*	AFAA	CSF20470	12	0–20 cm	20200715-1-(2)_0–20 cm_A_R2_SC1	S.F. Chen, L.L. Liu, J.L. Han, L.S. Sun, and W.W. Li	OK167615	OK168652	OK169126	OK169210
*C. hongkongensis*	AFAA	CSF21233	90	0–20 cm	20200815-1-(3)_0–20 cm_B_R2_SC2	L.L. Liu, J.L. Han, and L.S. Sun	OK167629	OK168666	OK169127	OK169211
*C. hongkongensis*	AGAA	CSF20380	9	0–20 cm	20200711-1-(6)_0–20 cm_B_R1_SC1	S.F. Chen, L.L. Liu, J.L. Han, Y Liu, and X.Y. Liang	OK167630	OK168667	OK169128	OK169212
*C. hongkongensis*	AGAA	CSF20441	11	0–20 cm	20200715-1-(1)_0–20 cm_A_R1_SC2	S.F. Chen, L.L. Liu, J.L. Han, L.S. Sun, and W.W. Li	OK167631	OK168668	OK169129	OK169213
*C. hongkongensis*	AGAA	CSF20528	17	40–60 cm	20200715-1-(7)_0–60 cm_A_R1_SC1	S.F. Chen, L.L. Liu, J.L. Han, L.S. Sun, and W.W. Li	OK167632	OK168669	OK169130	OK169214
*C. hongkongensis*	AGAA	CSF21018	71	0–20 cm	20200811-1-(7)_0–20 cm_B_R1_SC1	L.L. Liu, J.L. Han, and L.S. Sun	OK167644	OK168681	OK169131	OK169215
*C. hongkongensis*	AHAA	CSF20760	52	0–20 cm	20200809-1-(2)_0–20 cm_B_R1_SC1	L.L. Liu, J.L. Han, and L.S. Sun	OK167645	OK168682	OK169132	OK169216
*C. hongkongensis*	AHAA	CSF20761 ^6^	52	0–20 cm	20200809-1-(2)_0–20 cm_B_R1_SC2	L.L. Liu, J.L. Han, and L.S. Sun	OK167646	OK168683	OK169133	OK169217
*C. hongkongensis*	AHAA	CSF21155	82	0–20 cm	20200813-1-(2)_0–20 cm_B_R2_SC1	L.L. Liu, J.L. Han, and L.S. Sun	OK167647	OK168684	OK169134	OK169218
*C. hongkongensis*	AHAA	CSF21156	82	0–20 cm	20200813-1-(2)_0–20 cm_B_R2_SC2	L.L. Liu, J.L. Han, and L.S. Sun	OK167648	OK168685	OK169135	OK169219
*C. hongkongensis*	BAAA	CSF20472	12	0–20 cm	20200715-1-(2)_0–20 cm_B_R1_SC1	S.F. Chen, L.L. Liu, J.L. Han, L.S. Sun, and W.W. Li	OK167649	OK168686	OK169136	OK169220
*C. hongkongensis*	BAAA	CSF20734	51	0–20 cm	20200809-1-(1)_0–20 cm_A_R1_SC1	L.L. Liu, J.L. Han, and L.S. Sun	OK167652	OK168689	OK169137	OK169221
*C. hongkongensis*	BAAA	CSF21183	86	0–20 cm	20200814-1-(2)_0–20 cm_B_R1_SC1	L.L. Liu, J.L. Han, and L.S. Sun	OK167657	OK168694	OK169138	OK169222
*C. hongkongensis*	BAAA	CSF21359	99	0–20 cm	20200816-1-(7)_0–20 cm_A_R2_SC1	L.L. Liu, J.L. Han, and L.S. Sun	OK167660	OK168697	OK169139	OK169223
*C. hongkongensis*	CAAA	CSF20353 ^6^	7	0–20 cm	20200711-1-(4)_0–20 cm_A_R2_SC2	S.F. Chen, L.L. Liu, J.L. Han, Y Liu, and X.Y. Liang	OK167664	OK168701	OK169140	OK169224
*C. hongkongensis*	CAAA	CSF20358	7	20–40 cm	20200711-1-(4)_0–40 cm_B_R1_SC1	S.F. Chen, L.L. Liu, J.L. Han, Y Liu, and X.Y. Liang	OK167665	OK168702	OK169141	OK169225
*C. hongkongensis*	CAAA	CSF20359	7	20–40 cm	20200711-1-(4)_0–40 cm_B_R1_SC2	S.F. Chen, L.L. Liu, J.L. Han, Y Liu, and X.Y. Liang	OK167666	OK168703	OK169142	OK169226
*C. hongkongensis*	CAAA	CSF20360 ^6^	7	20–40 cm	20200711-1-(4)_0–40 cm_B_R1_SC3	S.F. Chen, L.L. Liu, J.L. Han, Y Liu, and X.Y. Liang	OK167667	OK168704	OK169143	OK169227
*C. hongkongensis*	DAAA	CSF20334	6	20–40 cm	20200711-1-(3)_0–40 cm_B_R1_SC1	S.F. Chen, L.L. Liu, J.L. Han, Y Liu, and X.Y. Liang	OK167669	OK168706	OK169144	OK169228
*C. hongkongensis*	DAAA	CSF20383 ^6^	9	0–20 cm	20200711-1-(6)_0–20 cm_B_R2_SC2	S.F. Chen, L.L. Liu, J.L. Han, Y Liu, and X.Y. Liang	OK167673	OK168710	OK169145	OK169229
*C. hongkongensis*	DAAA	CSF20444	11	0–20 cm	20200715-1-(1)_0–20 cm_B_R1_SC1	S.F. Chen, L.L. Liu, J.L. Han, L.S. Sun, and W.W. Li	OK167678	OK168715	OK169146	OK169230
*C. hongkongensis*	DAAA	CSF21367	100	0–20 cm	20200816-1-(8)_0–20 cm_A_R1_SC1	L.L. Liu, J.L. Han, and L.S. Sun	OK167699	OK168736	OK169147	OK169231
*C. ilicicola*	AAAB	CSF20594	29	0–20 cm	20200727-1-(5)_0–20 cm_A_R2_SC1	L.L. Liu, J.L. Han, and L.S. Sun	OK168008	OK169045	OK169172	OK169256
*C. ilicicola*	AAAB	CSF21126	80	20–40 cm	20200812-1-(8)_0–40 cm_A_R2_SC1	L.L. Liu, J.L. Han, and L.S. Sun	OK168010	OK169047	OK169173	OK169257
*C. ilicicola*	AAAB	CSF21219	89	0–20 cm	20200815-1-(2)_0–20 cm_A_R2_SC1	L.L. Liu, J.L. Han, and L.S. Sun	OK168014	OK169051	OK169174	OK169258
*C. ilicicola*	AAAB	CSF21310 ^6^	96	20–40 cm	20200816-1-(4)_0–40 cm_A_R1_SC1	L.L. Liu, J.L. Han, and L.S. Sun	OK168016	OK169053	OK169175	OK169259
*C. ilicicola*	ABAA	CSF20618 ^6^	32	0–20 cm	20200729-1-(2)_0–20 cm_A_R1_SC1	L.L. Liu, J.L. Han, L.S. Sun, Y Liu, and X.Y. Liang	OK168034	OK169071	OK169176	OK169260
*C. ilicicola*	ABAA	CSF20620	32	0–20 cm	20200729-1-(2)_0–20 cm_A_R2_SC1	L.L. Liu, J.L. Han, L.S. Sun, Y Liu, and X.Y. Liang	OK168036	OK169073	OK169177	OK169261
*C. ilicicola*	ABAA	CSF20624	32	20–40 cm	20200729-1-(2)_0–40 cm_A_R1_SC1	L.L. Liu, J.L. Han, L.S. Sun, Y Liu, and X.Y. Liang	OK168038	OK169075	OK169178	OK169262
*C. ilicicola*	ABAA	CSF20703	45	0–20 cm	20200731-1-(2)_0–20 cm_B_R1_SC1	L.L. Liu, J.L. Han, and L.S. Sun	OK168042	OK169079	OK169179	OK169263
*C. ilicicola*	BBAA	CSF20853	61	20–40 cm	20200810-1-(5)_0–40 cm_A_R1_SC8	L.L. Liu, J.L. Han, and L.S. Sun	OK168043	OK169080	OK169180	OK169264
*C. ilicicola*	BBBA	CSF21052 ^6^	74	0–20 cm	20200812-1-(2)_0–20 cm_A_R1_SC1	L.L. Liu, J.L. Han, and L.S. Sun	OK168044	OK169081	OK169181	OK169265
*C. ilicicola*	BBBA	CSF21198	87	0–20 cm	20200814-1-(3)_0–20 cm_A_R2_SC2	L.L. Liu, J.L. Han, and L.S. Sun	OK168047	OK169084	OK169182	OK169266
*C. ilicicola*	BBBA	CSF21292	95	0–20 cm	20200816-1-(3)_0–20 cm_A_R1_SC1	L.L. Liu, J.L. Han, and L.S. Sun	OK168053	OK169090	OK169183	OK169267
*C. kyotensis*	AAAA	CSF20372	8	0–20 cm	20200711-1-(5)_0–20 cm_B_R2_SC1	S.F. Chen, L.L. Liu, J.L. Han, Y Liu, and X.Y. Liang	OK167950	OK168987	OK169160	OK169244
*C. kyotensis*	AAAA	CSF20443	11	0–20 cm	20200715-1-(1)_0–20 cm_A_R2_SC2	S.F. Chen, L.L. Liu, J.L. Han, L.S. Sun, and W.W. Li	OK167952	OK168989	OK169161	OK169245
*C. kyotensis*	AAAA	CSF21350	98	0–20 cm	20200816-1-(6)_0–20 cm_B_R1_SC1	L.L. Liu, J.L. Han, and L.S. Sun	OK167981	OK169018	OK169163	OK169247
*C. kyotensis*	AAAB	CSF20518	16	0–20 cm	20200715-1-(6)_0–20 cm_B_R2_SC1	S.F. Chen, L.L. Liu, J.L. Han, L.S. Sun, and W.W. Li	OK167953	OK168990	OK169162	OK169246
*C. kyotensis*	ABAA	CSF21191 ^6^	86	40–60 cm	20200814-1-(2)_0–60 cm_B_R2_SC1	L.L. Liu, J.L. Han, and L.S. Sun	OK167998	OK169035	OK169167	OK169251
*C. kyotensis*	ABAB	CSF20260	1	0–20 cm	20200709-1-(1)_0–20 cm_A_R2_SC2	S.F. Chen, L.L. Liu, J.L. Han, Y Liu, and X.Y. Liang	OK167983	OK169020	OK169164	OK169248
*C. kyotensis*	ABAB	CSF20432	10	40–60 cm	20200711-1-(7)_0–60 cm_B_R2_SC1	S.F. Chen, L.L. Liu, J.L. Han, Y Liu, and X.Y. Liang	OK167988	OK169025	OK169166	OK169250
*C. kyotensis*	ABBA	CSF20338	6	20–40 cm	20200711-1-(3)_0–40 cm_B_R2_SC1	S.F. Chen, L.L. Liu, J.L. Han, Y Liu, and X.Y. Liang	OK167986	OK169023	OK169165	OK169249
*C. kyotensis*	BAAA	CSF20275	2	20–40 cm	20200709-1-(2)_0–40 cm_A_R1_SC1	S.F. Chen, L.L. Liu, J.L. Han, Y Liu, and X.Y. Liang	OK168002	OK169039	OK169168	OK169252
*C. kyotensis*	BAAA	CSF20276 ^6^	2	20–40 cm	20200709-1-(2)_0–40 cm_A_R1_SC2	S.F. Chen, L.L. Liu, J.L. Han, Y Liu, and X.Y. Liang	OK168003	OK169040	OK169169	OK169253
*C. kyotensis*	BAAA	CSF21111	78	0–20 cm	20200812-1-(6)_0–20 cm_B_R2_SC1	L.L. Liu, J.L. Han, and L.S. Sun	OK168006	OK169043	OK169170	OK169254
*C. kyotensis*	BAAA	CSF21335 ^6^	97	0–20 cm	20200816-1-(5)_0–20 cm_A_R1_SC2	L.L. Liu, J.L. Han, and L.S. Sun	OK168007	OK169044	OK169171	OK169255
*C. orientalis*	AAAA	CSF20602	31	40–60 cm	20200729-1-(1)_0–60 cm_A_R1_SC1	L.L. Liu, J.L. Han, L.S. Sun, Y Liu, and X.Y. Liang	OK168057	OK169094	OK169186	OK169270
*C. orientalis*	AAAA	CSF20603	31	40–60 cm	20200729-1-(1)_0–60 cm_A_R1_SC2	L.L. Liu, J.L. Han, L.S. Sun, Y Liu, and X.Y. Liang	OK168058	OK169095	OK169187	OK169271
*C. orientalis*	AAAA	CSF20606	31	40–60 cm	20200729-1-(1)_0–60 cm_B_R1_SC1	L.L. Liu, J.L. Han, L.S. Sun, Y Liu, and X.Y. Liang	OK168061	OK169098	OK169188	OK169272
*C. orientalis*	AAAA	CSF20607	31	40–60 cm	20200729-1-(1)_0–60 cm_B_R1_SC2	L.L. Liu, J.L. Han, L.S. Sun, Y Liu, and X.Y. Liang	OK168062	OK169099	OK169189	OK169273
*C. orientalis*	AAAA	CSF20610	31	40–60 cm	20200729-1-(1)_0–60 cm_B_R2_SC1	L.L. Liu, J.L. Han, L.S. Sun, Y Liu, and X.Y. Liang	OK168064	OK169101	OK169190	OK169274
*C. orientalis*	AAAA	CSF20611	31	40–60 cm	20200729-1-(1)_0–60 cm_B_R2_SC2	L.L. Liu, J.L. Han, L.S. Sun, Y Liu, and X.Y. Liang	OK168065	OK169102	OK169191	OK169275
*C. orientalis*	AAAA	CSF20614 ^6^	31	60–80 cm	20200729-1-(1)_0–80 cm_B_R1_SC1	L.L. Liu, J.L. Han, L.S. Sun, Y Liu, and X.Y. Liang	OK168068	OK169105	OK169192	OK169276
*C. orientalis*	AAAA	CSF20615	31	60–80 cm	20200729-1-(1)_0–80 cm_B_R1_SC2	L.L. Liu, J.L. Han, L.S. Sun, Y Liu, and X.Y. Liang	OK168069	OK169106	OK169193	OK169277

^1^ Genotype within each *Calonectria* species, determined by sequences of the *tef1*, *tub2*, *cmdA,* and *his3* regions; “-” means not available. ^2^ CSF: Culture collection located at China Eucalypt Research Centre (CERC), Chinese Academy of Forestry, ZhanJiang, GuangDong Province, China. ^3^ Number of 100 sampling points in this study. ^4^ Information associated with sample point and isolate, for example, “20200711-1-(3)_0–20 cm_A_R2_SC2” indicated sample number “20200711-1-(3), soil layer (0–20 cm), sample plastic bag (A), plastic sampling cup (R2), single conidium 2 (SC2). ^5^
*tef1* = translation elongation factor 1-alpha; *tub2* = β-tubulin; *cmdA* = calmodulin; *his3* = histone H3. ^6^ Isolates used for measuring macroconidia and vesicles in the current study. ^7^ “–” represents the relative locus was not successfully amplified in the current study.

**Table 4 jof-07-00857-t004:** Isolates from other studies used in the phylogenetic analyses in this study.

Species Code ^1^	Species	Isolates No. ^2,3^	Other Collection Number ^3^	Hosts	Area of Occurrence	Collector	GenBank Accession No. ^4^				References
							*tef1*	*tub2*	*cmdA*	*his3*	
Species in *Calonectria kyotensis* species complex											
B4	*C. aconidialis*	CMW 35174^T^	CBS 136086; CERC 1850	Soil in *Eucalyptus* plantation	HaiNan, China	X. Mou and S.F. Chen	MT412695	**OK357463**	MT335165	MT335404	[5,9]
		CMW 35384	CBS 136091; CERC 1886	Soil in *Eucalyptus* plantation	HaiNan, China	X. Mou and S.F. Chen	MT412696	**OK357464**	MT335166	MT335405	[5,9]
B5	*C. aeknauliensis*	CMW 48253^T^	CBS 143559	Soil in *Eucalyptus* plantation	Aek Nauli, North Sumatra, Indonesia	M.J. Wingfield	MT412710	**OK357465**	MT335180	MT335419	[5,12]
		CMW 48254	CBS 143560	Soil in *Eucalyptus* plantation	Aek Nauli, North Sumatra, Indonesia	M.J. Wingfield	MT412711	**OK357466**	MT335181	MT335420	[5,12]
B8	*C. asiatica*	CBS 114073^T^	CMW 23782; CPC 3900	Debris leaf litter	Prathet Thai, Thailand	N.L. Hywel-Jones	AY725705	AY725616	AY725741	AY725658	[29,49]
B17	*C. brassicicola*	CBS 112841^T^	CMW 51206; CPC 4552	Soil at *Brassica* sp.	Indonesia	M.J. Wingfield	KX784689	KX784619	KX784561	N/A ^5^	[30]
B19	*C. bumicola*	CMW 48257^T^	CBS 143575	Soil in *Eucalyptus* plantation	Aek Nauli, North Sumatra, Indonesia	M.J. Wingfield	MT412736	**OK357467**	MT335205	MT335445	[5,12]
B20	*C. canadiana*	CMW 23673^T^	CBS 110817; STE-U 499	*Picea* sp.	Canada	S. Greifenhagen	MT412737	MT412958	MT335206	MT335446	[1,5,17,50]
		CERC 8952	–	Soil	HeNan, China	S.F. Chen	MT412821	MT413035	MT335290	MT335530	[5,36]
B23	*C. chinensis*	CMW 23674^T^	CBS 114827; CPC 4101	Soil	Hong Kong, China	E.C.Y. Liew	MT412751	MT412972	MT335220	MT335460	[5,29,49]
		CMW 30986	CBS 112744; CPC 4104	Soil	Hong Kong, China	E.C.Y. Liew	MT412752	MT412973	MT335221	MT335461	[5,29,49]
B26	*C. cochinchinensis*	CMW 49915^T^	CBS 143567	Soil in *Hevea brasiliensis* plantation	Duong Minh Chau, Tay Ninh, Vietnam	N.Q. Pham, Q.N. Dang, and T.Q. Pham	MT412756	MT412977	MT335225	MT335465	[5,12]
		CMW 47186	CBS 143568	Soil in *Acacia auriculiformis* plantation	Song May, Dong Nai, Vietnam	N.Q. Pham and T.Q. Pham	MT412757	MT412978	MT335226	MT335466	[5,12]
B29	*C. colombiensis*	CMW 23676^T^	CBS 112220; CPC 723	Soil in *E. grandis* trees	La Selva, Colombia	M.J. Wingfield	MT412759	MT412980	MT335228	MT335468	[5,49]
		CMW 30985	CBS 112221; CPC 724	Soil in *E. grandis* trees	La Selva, Colombia	M.J. Wingfield	MT412760	MT412981	MT335229	MT335469	[5,49]
B31	*C. curvispora*	CMW 23693^T^	CBS 116159; CPC 765	Soil	Tamatave, Madagascar	P.W. Crous	MT412763	**OK357468**	MT335232	MT335472	[1,5,9,29,51]
		CMW 48245	CBS 143565	Soil in *Eucalyptus* plantation	Aek Nauli, North Sumatra, Indonesia	M.J. Wingfield	MT412764	N/A	MT335233	MT335473	[5,12]
B46	*C. heveicola*	CMW 49913^T^	CBS 143570	Soil in *H. brasiliensis* plantation	Bau Bang, Binh Duong, Vietnam	N.Q. Pham, Q.N. Dang, and T.Q. Pham	MT412786	MT413004	MT335255	MT335495	[5,12]
		CMW 49928	CBS 143571	Soil	Bu Gia Map National Park, Binh Phuoc, Vietnam	N.Q. Pham, Q.N. Dang, and T.Q. Pham	MT412811	MT413025	MT335280	MT335520	[5,12]
B48	*C. hongkongensis*	CBS 114828^T^	CMW 51217; CPC 4670	Soil	Hong Kong, China	M.J. Wingfield	MT412789	MT413007	MT335258	MT335498	[5,49]
		CERC 3570	CMW 47271	Soil in *Eucalyptus* plantation	BeiHai, Guangxi, China	S.F. Chen, J.Q. Li, and G.Q. Li	MT412791	MT413009	MT335260	MT335500	[5,11]
B51	*C. ilicicola*	CMW 30998^T^	CBS 190.50; IMI 299389; STE-U 2482	*Solanum tuberosum*	Bogor, Java, Indonesia	K.B. Boedijn and J. Reitsma	MT412797	**OK357469**	MT335266	MT335506	[1,5,29,52]
B52	*C. indonesiae*	CMW 23683^T^	CBS 112823; CPC 4508	*Syzygium aromaticum*	Warambunga, Indonesia	M.J. Wingfield	MT412798	MT413015	MT335267	MT335507	[5,49]
		CBS 112840	CMW 51205; CPC 4554	*S. aromaticum*	Warambunga, Indonesia	M.J. Wingfield	MT412799	MT413016	MT335268	MT335508	[5,49]
B55	*C. kyotensis*	CBS 114525^T^	ATCC 18834; CMW 51824; CPC 2367	*Robinia pseudoacacia*	Japan	T. Terashita	MT412802	MT413019	MT335271	MT335511	[1,5,30,53]
		CBS 114550	CMW 51825; CPC 2351	Soil	China	M.J. Wingfield	MT412777	MT412995	MT335246	MT335486	[5,30]
B57	*C. lantauensis*	CERC 3302^T^	CBS 142888; CMW 47252	Soil	LiDao, Hong Kong, China	M.J. Wingfield and S.F. Chen	MT412803	**OK357470**	MT335272	MT335512	[5,11]
		CERC 3301	CBS 142887; CMW 47251	Soil	LiDao, Hong Kong, China	M.J. Wingfield and S.F. Chen	MT412804	**OK357471**	MT335273	MT335513	[5,11]
B58	*C. lateralis*	CMW 31412^T^	CBS 136629	Soil in *Eucalyptus* plantation	GuangXi, China	X. Zhou, G. Zhao, and F. Han	MT412805	MT413020	MT335274	MT335514	[5,9]
B66	*C. malesiana*	CMW 23687^T^	CBS 112752; CPC 4223	Soil	Northern Sumatra, Indonesia	M.J. Wingfield	MT412817	MT413031	MT335286	MT335526	[5,49]
		CBS 112710	CMW 51199; CPC 3899	Leaf litter	Prathet, Thailand	N.L. Hywel-Jones	MT412818	MT413032	MT335287	MT335527	[5,49]
B80	*C. pacifica*	CMW 16726^T^	A1568; CBS 109063; IMI 354528; STE-U 2534	*Araucaria heterophylla*	Hawaii, USA	M. Aragaki	MT412842	**OK357472**	MT335311	MT335551	[1,5,49,50]
		CMW 30988	CBS 114038	*Ipomoea aquatica*	Auckland, New Zealand	C.F. Hill	MT412843	**OK357473**	MT335312	MT335552	[1,5,29,49]
B86	*C. penicilloides*	CMW 23696^T^	CBS 174.55; STE-U 2388	*Prunus* sp.	Hatizyo Island, Japan	M. Ookubu	MT412869	MT413081	MT335338	MT335578	[1,5,54]
B112	*C. sumatrensis*	CMW 23698^T^	CBS 112829; CPC 4518	Soil	Northern Sumatra, Indonesia	M.J. Wingfield	MT412913	**OK357474**	MT335382	MT335622	[5,49]
		CMW 30987	CBS 112934; CPC 4516	Soil	Northern Sumatra, Indonesia	M.J. Wingfield	MT412914	**OK357475**	MT335383	MT335623	[5,49]
B113	*C. syzygiicola*	CBS 112831^T^	CMW 51204; CPC 4511	*Syzygium aromaticum*	Sumatra, Indonesia	M.J. Wingfield	KX784736	KX784663	N/A	N/A	[30]
B116	*C. uniseptata*	CBS 413.67^T^	CMW 23678; CPC 2391; IMI 299577	*Paphiopedilum callosum*	Celle, Germany	W. Gerlach	GQ267307	GQ267208	GQ267379	GQ267248	[30]
B120	*C. yunnanensis*	CERC 5339^T^	CBS 142897; CMW 47644	Soil in *Eucalyptus* plantation	YunNan, China	S.F. Chen and J.Q. Li	MT412927	MT413134	MT335396	MT335636	[5,11]
		CERC 5337	CBS 142895; CMW 47642	Soil in *Eucalyptus* plantation	YunNan, China	S.F. Chen and J.Q. Li	MT412928	MT413135	MT335397	MT335637	[5,11]
B124	*C. singaporensis*	CBS 146715^T^	MUCL 048320	leaf litter submerged in a small stream	Mac Ritchie Reservoir, Singapore	C. Decock	MW890086	MW890124	MW890042	MW890055	[34]
		CBS 146713	MUCL 048171	leaf litter submerged in a small stream	Mac Ritchie Reservoir, Singapore	C. Decock	MW890084	MW890123	MW890040	MW890053	[34]
Species in *Calonectria brassicae* species complex											
B12	*C. brachiatica*	CMW 25298^T^	CBS 123700	*Pinus maximinoi*	Buga, Colombia	M.J. Wingfield	MT412726	MT412948	MT335195	MT335435	[5,7]
		CMW 25302	–	*Pinus ecunumanii*	Buga, Colombia	M.J. Wingfield	MT412727	MT412949	MT335196	MT335436	[5,7]
B16	*C. brassicae*	CBS 111869^T^	CPC 2409	*Argyreia splendens*	Indonesia	F. Bugnicourt	MT412733	MT412955	MT335202	MT335442	[1,5,29,30]
B25	*C. clavata*	CMW 23690^T^	ATCC 66389; CBS 114557; CPC 2536; P078-1543	*Callistemon viminalis*	Lake Placid, Florida, USA	C.P. Seymour and E.L. Barnard	MT412754	MT412975	MT335223	MT335463	[1,5,29,55]
		CMW 30994	CBS 114666; CPC 2537; P078-1261	Root debris in peat	Lee County, Florida, USA	D. Ferrin	MT412755	MT412976	MT335224	MT335464	[1,5,29,55]
B34	*C. duoramosa*	CBS 134656^T^	–	Soil in tropical rainforest	Monte Dourado, Pará, Brazil	R.F. Alfenas	KM395853	KM395940	KM396027	KM396110	[10]
		LPF453	–	Soil in *Eucalyptus* plantation	Monte Dourado, Pará, Brazil	R.F. Alfenas	KM395854	KM395941	KM396028	KM396111	[10]
B35	*C. ecuadorae*	CMW 23677^T^	CBS 111406; CPC 1635	Soil	Ecuador	M.J. Wingfield	MT412773	MT412991	MT335242	MT335482	[5,29,56]
		CBS 111706	CMW 51821; CPC 1636	Soil	Ecuador	M.J. Wingfield	MT412771	MT412989	MT335240	MT335480	[5,31]
B43	*C. gracilis*	CBS 111807^T^	AR2677; CMW 51189; STE-U 2634	*Manilkara zapota*	Pará, Brazil	F. Carneiro de Albuquerque	GQ267323	AF232858	GQ267407	DQ190646	[1,30,31,56,57]
		CBS 111284	CMW 51175; CPC 1483	Soil	Imbrapa, Brazil	P.W. Crous	GQ267324	DQ190567	GQ267408	DQ190647	[1,30,31,56,57]
B77	*C. octoramosa*	CBS 111423^T^	CMW 51819; CPC 1650	Soil	Ecuador	M.J. Wingfield	MT412834	MT413048	MT335303	MT335543	[5,31]
B78	*C. orientalis*	CMW 20291^T^	CBS 125260	Soil	Langam, Indonesia	M.J. Wingfield	MT412835	MT413049	MT335304	MT335544	[5,29]
		CMW 20273	CBS 125259	Soil	Teso East, Indonesia	M.J. Wingfield	MT412836	MT413050	MT335305	MT335545	[5,29]
B82	*C. paraensis*	CBS 134669^T^	LPF430	Soil in *Eucalyptus* plantation	Monte Dourado, Pará, Brazil	R.F. Alfenas	KM395837	KM395924	KM396011	KM396094	[10]
		LPF429	–	Soil in tropical rainforest	Monte Dourado, Pará, Brazil	R.F. Alfenas	KM395841	KM395928	KM396015	KM396098	[10]
B83	*C. parvispora*	CBS 111465^T^	CPC 1902	Soil	Brazil	A.C. Alfenas	MT412845	MT413057	MT335314	MT335554	[5,31]
		CMW 30981	CBS 111478; CPC 1921	Soil	Brazil	A.C. Alfenas	MT412844	MT413056	MT335313	MT335553	[5,29,30]
B84	*C. pauciphialidica*	CMW 30980^T^	CBS 111394; CPC 1628	Soil	Ecuador	M.J. Wingfield	MT412846	MT413058	MT335315	MT335555	[5,29,56]
B88	*C. pini*	CMW 31209^T^	CBS 123698	*Pinus patula*	Buga, Valle del Cauca, Colombia	C.A. Rodas	MT412870	MT413082	MT335339	MT335579	[5,29]
		CBS 125523	CMW 31210	*Pinus patula*	Buga, Valle del Cauca, Colombia	C.A. Rodas	GQ267345	GQ267225	GQ267437	GQ267274	[29]
B91	*C. pseudobrassicae*	CBS 134662^T^	LPF280	Soil in *Eucalyptus* plantation	Santana, Pará, Brazil	A.C. Alfenas	KM395849	KM395936	KM396023	KM396106	[10]
		CBS 134661	LPF260	Soil in *Eucalyptus* plantation	Santana, Pará, Brazil	A.C. Alfenas	KM395848	KM395935	KM396022	KM396105	[10]
B92	*C. pseudoecuadoriae*	CBS 111402^T^	CMW 51179; CPC 1639	Soil	Ecuador	M.J. Wingfield	KX784723	KX784652	KX784589	N/A	[30,31]
B105	*C. quinqueramosa*	CBS 134654^T^	LPF065	Soil in *Eucalyptus* plantation	Monte Dourado, Pará, Brazil	R.F. Alfenas	KM395855	KM395942	KM396029	KM396112	[10]
		CBS 134655	LPF281	Soil in *Eucalyptus* plantation	Santana, Pará, Brazil	A.C. Alfenas	KM395856	KM395943	KM396030	KM396113	[10]
B107	*C. robigophila*	CBS 134652^T^	LPF192	*Eucalyptus* sp. leaf	Açailandia, Maranhao, Brazil	R.F. Alfenas	KM395850	KM395937	KM396024	KM396107	[10]
		CBS 134653	LPF193	*Eucalyptus* sp. leaf	Açailandia, Maranhao, Brazil	R.F. Alfenas	KM395851	KM395938	KM396025	KM396108	[10]
Outgroups											
	*Curvicladiella cignea*	CBS 109167^T^	CPC 1595; MUCL 40269	Decaying leaf	French Guiana	C. Decock	KM231867	KM232002	KM231287	KM231461	[56,58,59]
		CBS 109168	CPC 1594; MUCL 40268	Decaying seed	French Guiana	C. Decock	KM231868	KM232003	KM231286	KM231460	[56,58,59]

^1^ Codes B1 to B120 of the 120 accepted *Calonectria* species resulting from Liu and co-authors [5], “B124” indicated *C. singaporensis* described in Crous and co-authors [34]. ^2^ T: ex-type isolates of the species. ^3^ AR: Amy Y. Rossman working collection; ATCC: American Type Culture Collection, Virginia, USA; CBS: Westerdijk Fungal Biodiversity Institute, Utrecht, The Netherlands; CERC: China Eucalypt Research Centre, ZhanJiang, GuangDong Province, China; CMW: Culture collection of the Forestry and Agricultural Biotechnology Institute FABI, University of Pretoria, Pretoria, South Africa; CPC: Pedro Crous working collection housed at Westerdijk Fungal Biodiversity Institute; IMI: International Mycological Institute, MUCL: Mycotheque, Laboratoire de Mycologie Systematique st Appliqee, I’Universite, Louvian-la-Neuve, Belgium; STE-U: Department of Plant Pathology, University of Stellenbosch, South Africa; “–” represents no other collection number. ^4^
*tef1*: translation elongation factor 1-alpha; *tub2*: β-tubulin; *cmdA*: calmodulin; *his3*: histone H3; for GenBank Accession No. in bold, the sequences were submitted in this study. ^5^ N/A represents data that is not available.

**Table 5 jof-07-00857-t005:** Statistical values of datasets for maximum parsimony and maximum likelihood analyses in this study.

Dataset	No. of Taxa	No. of bp ^1^	Maximum Parsimony						
			PIC ^2^	No. of Trees	Tree Length	CI ^3^	RI ^4^	RC ^5^	HI ^6^
*tef1*	157	522	241	110	588	0.697	0.973	0.678	0.303
*tub2*	156	597	256	1000	694	0.635	0.967	0.614	0.365
*cmdA*	156	697	277	1000	617	0.676	0.969	0.655	0.324
*his3*	153	478	166	973	602	0.570	0.960	0.547	0.430
*tef1/tub2/cmdA/his3*	157	2303	944	150	2671	0.609	0.962	0.586	0.391
**Dataset**	**Maximum likelihood**								
	**Subst. mode ^7^**	**NST ^8^**	**Rate matrix**					**Rates**	
*tef1*	TIM2+G	6	1.8670	3.4436	1.8670	1.0000	5.0336	Gamma	
*tub2*	TPM3uf+I+G	6	1.4137	4.7965	1.0000	1.4137	4.7965	Gamma	
*cmdA*	TrN+G	6	1.0000	3.5934	1.0000	1.0000	7.2024	Gamma	
*his3*	GTR+I+G	6	2.5191	8.8466	5.6820	2.1055	15.5239	Gamma	
*tef1/tub2/cmdA/his3*	GTR+I+G	6	1.5966	4.2868	1.3927	0.9904	5.5003	Gamma	

^1^ bp = base pairs. ^2^ PIC = number of parsimony informative characters. ^3^ CI = consistency index. ^4^ RI = retention index. ^5^ RC = rescaled consistency index. ^6^ HI = homoplasy index. ^7^ Subst. model = best fit substitution model. ^8^ NST = number of substitution rate categories.

**Table 6 jof-07-00857-t006:** Morphological comparisons of *Calonectria* isolates and species obtained in the current study.

Species	Isolate/Species	Macroconidia (L × W) ^1,2,3^	Macroconidia Average (L × W) ^1,2^	Macroconidia Septation	Vesicle Width ^1,2,3^	Vesicle Width Average ^1^
*C. aconidialis*	Isolate CSF20323 (this study)	(35–)39.5–45.5(–48) × (4–)4–4.5(–5)	42.5 × 4.5	1	(3.5–)4.5–6(–6.5)	5
	Isolate CSF20376 (this study)	(34.5–)38.5–45(–47.5) × (4–)4.5–5(–5.5)	41.5 × 4.5	1	(4–)4.5–11(–13)	8
	Isolate CSF20985 (this study)	(41–)46.5–51.5(–54) × (4–)4.5–5(–5.5)	49 × 5	1	(4.5–)5–6.5(–9.5)	6
	Species (this study)	(34.5–)40–48.5(–54) × (4–)4.5–5(–5.5)	44.5 × 4.5	1	(3.5–)4–8.5(–13)	6
	Species [9]	N/A ^4^	N/A	N/A	N/A	N/A
*C. chinensis*	Isolate CSF20756 (this study)	(35.5–)40–45(–49) × (3.5–)4–4.5(–4.5)	42.5 × 4	1	(3.5–)3.5–9(–11.5)	6.5
	Isolate CSF20759 (this study)	(34.5–)37.5–43(–46) × (3.5–)4–4.5(–5)	40.5 × 4	1	(3–)5–10.5(–12)	8
	Species (this study)	(34.5–)38.5–44(–49) × (3.5–)4–4.5(–5)	41.5 × 4	1	(3–)4–10(–12)	7
	Species [49]	(38–)41–48(–56) × (3.5–)4(–4.5)	45 × 4	1	6–9	N/A
*C. hongkongensis*	Isolate CSF20353 (this study)	(33.5–)36–42(–48) × (3.5–)4–4.5(–4.5)	39 × 4	1	(4–)5–8.5(–10.5)	6.5
	Isolate CSF20360 (this study)	(34–)35.5–40(–43.5) × (3.5–)4–4.5(–5)	37.5 × 4	1	(4.5–)5.5–9(–11)	7.5
	Isolate CSF20383 (this study)	(37.5–)42.5–48(–50.5) × (4–)4–4.5(–5)	45.5 × 4.5	1	(4–)6–10.5(–11)	8.5
	Isolate CSF20761 (this study)	(32–)34.5–39.5(–43) × (3.5–)3.5–4(–4.5)	37 × 4	1	(4–)5.5–8(–9.5)	6.5
	Isolate CSF20924 (this study)	(35–)37.5–44(–45.5) × (3.5–)4–4.5(–5)	40.5 × 4	1	(6–)9–13(–14.5)	11
	Species (this study)	(32–)36–44(–50.5) × (3.5–)4–4.5(–5)	40 × 4	1	(4–)5.5–10.5(–14.5)	8
	Species [49]	(38–)45–48(–53) × 4(–4.5)	46.5 × 4	1	8–14	N/A
*C. ilicicola*	Isolate CSF20618 (this study)	(52.5–)56.5–66(–71.5) × (6–)6.5–7.5(–8)	61.5 × 7	1(–3)	(8–)9–11(–11.5)	10
	Isolate CSF21052 (this study)	(31–)50.5–69(–78) × (3–)5–7(–7.5)	59.5 × 6	1(–3)	(3.5–)5–8(–11)	6.5
	Isolate CSF21310 (this study)	(50–)55–62.5(–67) × (5.5–)6–7(–7.5)	58.5 × 6.5	(1–)3	(4–)6.5–10(–11.5)	8.5
	Species (this study)	(31–)53.5–66(–78) × (3–)6–7(–8)	60 × 6.5	1(–3)	(3.5–)6–10(–11.5)	8
	Species [1]	(45–)70–82(–90) × (4–)5–6.5(–7)	62 × 6	(1–)3	(6–)7–10(–12)	N/A
*C. kyotensis*	Isolate CSF20276 (this study)	(33.5–)36.5–44(–51) × (3.5–)4–4.5(–4.5)	40.5 × 4	1	(6.5–)8.5–11.5(–12.5)	10
	Isolate CSF21191(this study)	(29.5–)32.5–38.5(–42.5) × (3.5–)4–4.5(–5)	35.5 × 4	1	(5–)7.5–10.5(–11.5)	9
	Isolate CSF21335 (this study)	(32–)35.5–40(–43) × (3.5–)4–4.5(–5)	38 × 4	1	(5–)8–10(–11)	9
	Species (this study)	(29.5–)34.5–41.5(–51) × (3.5–)4–4.5(–5)	38 × 4	1	(5–)7.5–10.5(–12.5)	9
	Species [1]	(35–)45–50(–55) × 3–4(–5)	40 × 3.5	1	6–12	N/A
*C. orientalis*	Isolate CSF20614 (this study)	(30.5–)35–40(–43.5) × (4.5–)5–5.5(–5.5)	37.5 × 5	1	(3–)4–6.5(–7.5)	5
	Species [29]	(43–)46–50(–53) × 4(–5)	48 × 4	1	5–10	N/A

^1^ All of the measurements are in µm. Fifty macroconidia and vesicles were measured for each isolate, with the exception of the vesicle of isolate CSF20618, for which 25 vesicles were measured because of the limited number of vesicles produced. ^2^ L × W = length × width. ^3^ Measurements are presented in the format ((minimum–) (average—standard deviation)—(average + standard deviation) (–maximum)). ^4^ N/A represents data that are not available.

**Table 7 jof-07-00857-t007:** Isolate numbers of each genotype in each soil layer for each *Calonectria* species.

*Calonectria* Species	Soil Layer	Genotype Determined by *tef1* Gene Sequences	Number of Isolates Based on *tef1* Genotype	Genotype Determined by *tub2* Gene Sequence	Number of Isolates Based on *tub2* Genotype	Genotype Determined by *tef1* and *tub2* Gene Sequences	Number of Isolates Based on *tef1* and *tub2* Genotype	Number of Isolates in Each Soil Layer for Each Species
*C. hongkongensis*	0–20 cm	A	346	A	337	AA	310	373
		B	15	B	1	AB	1	
		C	1	C	4	AC	4	
		D	11	D	5	AD	5	
				E	2	AE	2	
				F	9	AF	9	
				G	11	AG	11	
				H	4	AH	4	
						BA	15	
						CA	1	
						DA	11	
	20–40 cm	A	186	A	197	AA	180	203
		C	4	D	2	AD	2	
		D	13	F	4	AF	4	
						CA	4	
						DA	13	
	40–60 cm	A	58	A	50	AA	47	61
		D	3	F	7	AF	7	
				G	4	AG	4	
						DA	3	
	60–80 cm	A	8	A	8	AA	8	8
	80–100 cm	A	16	A	20	AA	16	20
		D	4			DA	4	
*C. aconidialis*	0–20 cm	A	140	A	98	AA	98	140
				B	1	AB	1	
				C	41	AC	41	
	20–40 cm	A	74	A	40	AA	40	74
				B	8	AB	8	
				C	26	AC	26	
	40–60 cm	A	20	A	6	AA	6	20
				C	14	AC	14	
	60–80 cm	A	8	A	8	AA	8	8
	80–100 cm	A	8	A	4	AA	4	8
				C	4	AC	4	
*C. kyotensis*	0–20 cm	A	31	A	27	AA	25	33
		B	2	B	6	AB	6	
						BA	2	
	20–40 cm	A	10	A	12	AA	8	14
		B	4	B	2	AB	2	
						BA	4	
	40–60 cm	A	7	B	7	AB	7	7
	60–80 cm	A	4	B	4	AB	4	4
	80–100 cm	–	–	–	–	–	–	0
*C. ilicicola*	0–20 cm	A	9	A	4	AA	4	16
		B	7	B	12	AB	5	
						BB	7	
	20–40 cm	A	18	A	14	AA	14	19
		B	1	B	5	AB	4	
						BB	1	
	40–60 cm	A	8	A	8	AA	8	8
	60–80 cm	B	4	B	4	BB	4	4
	80–100 cm	–	–	–	–	–	–	0
*C. chinensis*	0–20 cm	A	2	A	2	AA	2	2
	20–40 cm	–	–	–	–	–	–	0
	40–60 cm	–	–	–	–	–	–	0
	60–80 cm	–	–	–	–	–	–	0
	80–100 cm	–	–	–	–	–	–	0
*C. orientalis*	0–20 cm	–	–	–	–	–	–	0
	20–40 cm	–	–	–	–	–	–	0
	40–60 cm	A	11	A	11	AA	11	11
	60–80 cm	A	4	A	4	AA	4	4
	80–100 cm	–	–	–	–	–	–	0

## Data Availability

Data is contained within the article and Appendix A.

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
