# Peer review of "Species Diversity and Distribution Characteristics of Calonectria in Five Soil Layers in a Eucalyptus Plantation"

_jof, 2021, doi:10.3390/jof7100857_

Round 1
Reviewer 1 Report
The manuscript presented for review contains valuable, original research material. A large amount of work translates into a great potential of the presented results. Unfortunately, the manuscript does not reflect this and should be redrafted. Some comments:
A well written abstract contains all the necessary information.
Keywords should be rewritten, do not include repetitions of the title.
The method of citation must be adapted to the requirements of the journal
Line 59-63 should be redrafted - the text is poorly written,
Why was Mega 7.0 used when the MEGA X version is available? How was the reliability of the tree checked?
I would suggest describing the attempts in a different way in the methodology to facilitate and simplify their description later in the work. Do not repeat 0-20cm etc. Describe I, II etc
Graph A Fig. 1 is not needed
Tab 1 and 2 - tab 1 should either be merged or deleted.
Tab. 4 is too long. You can first describe the samples from 1 soil layer, then separate the second layer with a line, I would remove the Collectors column - in the table I would put a link in the form of *, ** and the names of the collectors under the table.
Table 5 - Not needed. The information, ie sequence no., Host can be placed on the tree.
Line 321-325 isolate numbers are placed in the table and there it is enough not to repeat them in the text
Line 340-352 not clearly written, edit it
The tab. is too much and too big
The discussion is poor and does not include discussions with the results of other authors from other countries. In my opinion, this is the weakest part of the manuscrypt.
Author Response
Dear Reviewer One,
We appreciate the comments and suggestions from you that have helped us to improve the document.
Our point by point responses to your comments and details of the revisions have been listed and explained in the files “Oct., 6, 2021_Response to Reviewer One Comments and Changes Made.docx”.

Reviewer 2 Report
In this manuscript, the authors studied the species diversity and distribution characteristics of Calonectria in five soil layers in the Eucalyptus plantation. In this study, 1000 soil samples were collected from five soil layers (0–20 cm, 20–40 cm, 40–60 cm, 60–80 cm, and 80–100 cm) at 100 sampling points in one 15-year-old Eucalyptus urophylla hybrid plantation in southern China. In total, 1037 isolates of Calonectria present in all five soil layers were obtained from 93 of 100 sampling points. The 1037 isolates were identified based on DNA sequence comparisons of the translation elongation factor 1-alpha (tef1), β-tubulin (tub2), calmodulin (cmdA), and histone H3 (his3) gene regions as well as a combination of morphological characteristics. These isolates were identified as C. hongkongensis (665 isolates; 64.1%), C. aconidialis (250 isolates; 24.1%), C. kyotensis (58 isolates; 5.6%), C. ilicicola (47 isolates; 4.5%), C. chinensis (2 isolates; 0.2%), and C. orientalis (15 isolates; 1.5%). With the exception of C. orientalis, which resides in the C. brassicae species complex, the other five species all belonged to the C. kyotensis species complex. The results showed that the number of sampling points that yielded Calonectria and the number (and percentage) of Calonectria isolates obtained decreased with increasing depth of the soil. More than 84% of the isolates were obtained from the 0–20 cm and 20–40 cm soil layers. The deeper soil layers, presenting comparatively lower numbers, still harbored a considerable number of Calonectria. The diversity of five species in the C. kyotensis species complex decreased with increasing soil depth. The genotypes of isolates in each Calonectria species were determined by tef1 and tub2 gene sequences. For each species in the C. kyotensis species complex, in most cases, the number of genotypes decreased with increasing soil depth; the 0–20 cm soil layer contained all of the genotypes of each species. This study presents the first report of C. orientalis isolated in China. This species was isolated from 40–60 cm and 60–80 cm soil layers at only one sampling point, and only one genotype was present. This study will enhance reader understanding of the species diversity and distribution characteristics of Calonectria in different soil layers.
The manuscript is written very well, but I have found plagiarized material. Therefore, it should be cleaned before proceeding further for publication.
Change at L81 threats for management to threats to the management.
L84 long-term potential harm to potential long-term harm.
L405 present, in the to present in the.
I have found plagiarized lines in this manuscript at L9-10, L16, L19-22, L41-43, L57, L67-68, L76-80, L106-108, L124-125, L128-135, L140-146, L155-156, L166-168, L175, L178-179, L183-184, L188-189, L230-231, L233-234, L254-256, L270, L272-277, L282-283, L286-287, L289-290, L293-298, L303, L338-339, L444-445. Please clean it.
Author Response
Dear Reviewer Two,
We appreciate the comments and suggestions from you that have helped us to improve the document.
Our point by point responses to your comments and details of the revisions have been listed and explained in the files “Oct., 6, 2021_Response to Reviewer Two Comments and Changes Made.docx”.

Round 2
Reviewer 1 Report
The authors have adapted to my comments. Already in the previous version I wrote about the potential of results. Prefers to include the most up-to-date versions of programs in manuscripts. Although it gives the same results (answer 4). For readers who are just starting to work with the program, it may be a problem to find an older version. Congratulations on a job well done.
Reviewer 2 Report
I am happy with the author's comments and the Manuscript has improved a lot, hence can be accepted in its current form.